# Tangent Space Causal Inference: Leveraging Vector Fields for Causal Discovery in Dynamical Systems

**Kurt Butler**[*]    **Daniel Waxman**[*]    **Petar M. Djurić**
Department of Electrical and Computer Engineering
Stony Brook University
{ kurt.butler, daniel.waxman, petar.djuric} @stonybrook.edu

## Abstract

Causal discovery with time series data remains a challenging yet increasingly important task across many scientific domains. Convergent cross mapping (CCM) and related methods have been proposed to study time series that are generated by dynamical systems, where traditional approaches like Granger causality are unreliable. However, CCM often yields inaccurate results depending upon the quality of the data. We propose the Tangent Space Causal Inference (TSCI) method for detecting causalities in dynamical systems. TSCI works by considering vector fields as explicit representations of the systems' dynamics and checks for the degree of synchronization between the learned vector fields. The TSCI approach is model-agnostic and can be used as a drop-in replacement for CCM and its generalizations. We first present a basic version of the TSCI algorithm, which is shown to be more effective than the basic CCM algorithm with very little additional computation. We additionally present augmented versions of TSCI that leverage the expressive power of latent variable models and deep learning. We validate our theory on standard systems, and we demonstrate improved causal inference performance across a number of benchmark tasks.

## 1   Introduction

The discovery of causal relationships is one of the most fundamental goals of scientific work. When causal relationships are known and understood, we can explain the behavior of a system and understand how our actions or interventions upon the system will affect its behavior [27]. This kind of reasoning is fundamental in many problem domains, such as medicine, environmental policy, and economics. However, it is not always possible to perform interventions and observe their effects. For example, medical practitioners might only have one chance to prescribe a medication to help a patient. As another example, an ecologist might find that performing a large experiment is forbiddingly expensive or otherwise infeasible. Due to these concerns, there remains a great interest in the problem of *observational* causal inference, where one infers causation without manipulating the system under study directly.

For time series data, the most prominent tool for observational causal inference is Granger causality [15]. Granger causality operates on the assumption that if one system (the cause) is driving changes in another system (the effect), then the cause should have unique information about what will happen to the effect. This is encoded in an assumption called *separability*, which posits that the information needed to predict the future behavior of the effect is not contained in the effect itself. However, the separability condition runs counter to the behavior of coupled dynamical systems: if long histories of the effect can forecast the cause, then an autoregressive model with the appropriate lag may forecast the effect from itself. This effect is explored in the supplementary materials of [12, 41], and we give

---

[*]Equal contribution.

38th Conference on Neural Information Processing Systems (NeurIPS 2024).

an example in Appendix B. Since systems with (approximately) dynamical behavior are ubiquitous in many application domains, alternative methods have become of interest.

To address the failures of Granger causality in coupled dynamical systems Sugihara et al. [41] proposed the convergent cross mapping (CCM) method, which directly takes advantage of the topological properties of dynamical systems. CCM can be seen as an adaption of earlier work that studied the synchronization of dynamical systems [2, 29, 35] into an algorithmic procedure for detecting causalities between time series. Using Takens' theorem, a well-known result in dynamical systems theory [42], CCM attempts to detect causality by reconstructing the state space of a given time series, and then learning a time-invariant function that maps between reconstructed state spaces, called a cross map. The key assumption is that the cross map only exists if the original systems were dynamically coupled. In this sense, the "causality" of CCM deviates from the popular Pearlian framework [27], and is better interpreted as identifying which variables "drive" or "force" a dynamical system based on observational data.[2] Building upon this framework, several extensions of CCM have been proposed to address various technicalities and caveats of the approach and to generalize the method to new domains. However, the CCM test statistic is difficult to interpret and does not admit a simple decision rule. CCM also does not explicitly learn a cross map function.

To improve upon CCM, we propose Tangent Space Causal Inference (TSCI). TSCI detects causation between dynamical systems by explicitly checking if the dynamics of one system map on to the other. The proposed method can be seen as a drop-in replacement for the CCM test, providing an interpretable and principled alternative while remaining compatible with many of the extensions of CCM. Furthermore, the proposed method is model agnostic, meaning that it can be adapted to any method used to learn the cross map function, including multilayer perception (MLP) networks, splines, or Gaussian process regression. The only major assumption of TSCI is that the time series under study were generated by continuous time dynamical systems (i.e, by systems of differential equations), which is a standard assumption in a number of physical systems [36]. As a result, TSCI is applicable to many of the same problems as CCM.

## 1.1 Related Work

**Causal representation learning.** The primary function of Takens' theorem in the CCM method is to yield a representation of a system's latent state so that it may be used for cross mapping. As a result, numerous generalizations of Takens' theorem emerged in the following decades [32, 33, 40]. Causal representation learning aims to learn hidden variables from high dimensional observations [1], and a principle task is to decipher the causal relationships between many, possibly redundant, observations of a system [30]. In such cases, methods of dimensionality reduction can be applied to extract causal variables from the raw observations [31]. This is particularly important in the processing of large, spatio-temporal data sets [36, 43]. Some authors have proposed that CCM can be improved by aggregating data from multiple sources into the reconstruction of the latent states [7, 45], which requires an awareness of how the observation data relate to the causal variables of interest.

**Causal discovery with cross maps.** CCM has popularized the use of cross maps as a tool for causal inference, and many variations and improvements of the basic CCM methodology have been proposed. Some of these works aim to improve the reconstruction of latent states in these models by using a Bayesian approach to latent state inference [13], where the approach is adapted for spatial/geographic data [14] or modified for sporadically sampled time series [12]. Additionally, several improvements have suggested changing the way that a cross map is detected: some authors have recommended varying the library length [25] or time delaying the cross mapping [44] to yield refined information. The $k$-nearest neighbor regression, which is used in the original CCM algorithm, can be swapped out for a radial basis function network [23], or for a Gaussian process regression model [13]. Other approaches have not directly learned the cross map at all; instead they have examined other aspects of the reconstructions such as their dimensionality [3] or used pairwise distance rankings as a signature of the mapping [5].

---

[2]Although not typically interpreted as such, cross maps have been suggested to be interpretable in a Pearlian causal framework via the *do* operator [12].

## 2 Proposed Method

We begin by considering the inference of a causal relationship between two time series, $x(t)$ and $y(t)$. Our starting assumption is that both time series were generated by latent dynamical systems with states $\mathbf{z}_x(t)$ and $\mathbf{z}_y(t)$, respectively, whose behavior is governed by a set of ordinary differential equations (ODEs). For motivation, we consider a particular case with a unidirectional coupling between the latent states:

$$x(t) = h_x(\mathbf{z}_x(t)), \tag{1}$$

$$y(t) = h_y(\mathbf{z}_y(t)), \tag{2}$$

$$\frac{d\mathbf{z}_x}{dt} = \mathbf{f}_x(\mathbf{z}_x), \tag{3}$$

$$\frac{d\mathbf{z}_y}{dt} = \mathbf{f}_y(\mathbf{z}_x, \mathbf{z}_y), \tag{4}$$

where the observation functions $h_x, h_y$ and the time derivative functions $\mathbf{f}_x, \mathbf{f}_y$ are assumed to be differentiable. A causal relationship between $x$ and $y$ is evidenced by the appearance of $\mathbf{z}_x$ in the equation for $d\mathbf{z}_y/dt$. The state vectors $\mathbf{z}_x$ and $\mathbf{z}_y$ could possibly have redundant information, but by writing the system in this form, we obtain an asymmetry between $x(t)$ and $y(t)$ which will inform our inference of causation. In our notation, if $\mathbf{z}_x$ appears in the equation for $d\mathbf{z}_y/dt$, we will write $x \rightarrow y$.

We now explain how the CCM method approaches this problem and how the TSCI method builds upon the CCM framework.

### 2.1 Convergent Cross Mapping

CCM is a technique for inferring causation between time series generated by dynamical systems [41], as seen in Eqs. (1) to (4). The basic motivation for CCM is that given an observed time series $x(t)$, one can construct a vector $\tilde{\mathbf{x}}$ which acts as a proxy for the latent states $\mathbf{z}_x$ that generated it. Given two constructions, $\tilde{\mathbf{x}}$ and $\tilde{\mathbf{y}}$, we may detect if there is a mapping between them, which provides evidence of a causal relationship. From Eq. (4), since $\mathbf{z}_x$ influences $\mathbf{z}_y$ unidirectionally, the effect $y(t)$ contains more information than the cause time series $x(t)$, and as a result $\tilde{\mathbf{y}}$ generally contains enough information to reconstruct $\tilde{\mathbf{x}}$. With this in mind, we can frame CCM as a two-step procedure [13]. In Step 1, we construct representations $\tilde{\mathbf{x}}$ and $\tilde{\mathbf{y}}$ that are proxies for $\mathbf{z}_x$ and $\mathbf{z}_y$, respectively. In Step 2, we detect a mapping between reconstructions; if there is a mapping $\tilde{\mathbf{y}} \mapsto \tilde{\mathbf{x}}$, then the reverse causality holds, $x \rightarrow y$.

To reconstruct the latent state space, as in Step 1, multiple approaches could be employed. However, in the basic CCM methodology, one uses the so-called *delay embedding* of $x(t)$,

$$\tilde{\mathbf{x}}(t) = \begin{bmatrix} x(t) \\ x(t-\tau) \\ \vdots \\ x(t-(Q-1)\tau) \end{bmatrix}, \tag{5}$$

where $\tau$ and $Q$ are parameters called the embedding lag and embedding dimension, respectively. The justification that $\tilde{\mathbf{x}}$ is a good proxy for $\mathbf{z}_x$ is given by Takens' theorem [42], which states that $\tilde{\mathbf{x}}$ and $\mathbf{z}_x$ are equivalent up to a nonlinear change of coordinates.

**Theorem 2.1 (Takens' theorem [33])** *Let $M$ be a compact manifold of dimension $d$. Let $\mathbf{z} \in M$ evolve according to $d\mathbf{z}/dt = \mathbf{f}(\mathbf{z})$, let $\phi_\tau$ be the mapping that takes $\mathbf{z}_t$ to $\mathbf{z}_{t+\tau}$, and let $x(t) = h(\mathbf{z}(t))$. If $Q \geq 2d+1$, then for almost-every[3] triplet $(\mathbf{f}, h, \tau)$, the map $\Phi_{\mathbf{f},h,\tau}$*

$$\Phi_{\mathbf{f},h,\tau}(\mathbf{z}) = \begin{bmatrix} h(\mathbf{z}) \\ h(\phi_{-\tau}(\mathbf{z})) \\ \vdots \\ h(\phi_{-(Q-1)\tau}(\mathbf{z})) \end{bmatrix} \tag{6}$$

---

[3]Several version of Takens' theorem exist, and they make useful, but mathematically distinct, statements about how common such embeddings are. The version presented here says that functions that do *not* produce embeddings live on a measure zero set, in the sense of prevalence [33, p. 584].

*is an embedding of $M$ into $\mathbb{R}^Q$.*

Since $\Phi_{\mathbf{f},h,\tau}(\mathbf{z}_x(t)) = \tilde{\mathbf{x}}(t)$, Takens' theorem tells us that if $\mathbf{z}_x$ lives on a manifold $M$, then the points $\tilde{\mathbf{x}}$ lie on a manifold $\mathcal{M}_x$, which is called the *shadow manifold* of $x$ [41]. Despite the number of assumptions in the statement of Takens' theorem, many generalizations of the statement exist, allowing us to justify the use of delay embedding to systems with strange attractors [33] and systems with noise [40]. If a system satisfies the conditions for Takens' theorem, in the sense that $\mathcal{M}_x$ is a valid embedding of $M$, then we say that the system is *generic*.

Since only $\mathbf{z}_x$ influences $x(t)$, Takens' theorem implies that $\tilde{\mathbf{x}}$ is equivalent (up to nonlinear coordinate change) to $\mathbf{z}_x$. However, due to the appearance of $\mathbf{z}_x$ in (4), both $\mathbf{z}_x$ and $\mathbf{z}_y$ are responsible for generating $y(t)$. As a result, Takens' theorem suggests that $\tilde{\mathbf{y}}$ is equivalent to $(\mathbf{z}_x, \mathbf{z}_y)$ as a concatenated vector [39]. Since $(\mathbf{z}_x, \mathbf{z}_y)$ clearly can be mapped onto $\mathbf{z}_x$, the equivalence between the delay embeddings and the latent states suggests that there is a mapping $\tilde{\mathbf{y}} \mapsto \tilde{\mathbf{x}}$, called a *cross map*. Thus, cross maps encode the idea that the effect time series contains information about its cause. This is encoded in the following corollary to Takens' theorem.

**Corollary 2.1.1** *Suppose that $x \to y$ for a generic system. Then there exists a function $F$ such that $\tilde{\mathbf{x}}(t) = F(\tilde{\mathbf{y}}(t))$ for all $t$.*

The proof of the corollary is provided in the Appendix.

Step 2 of CCM is then to detect if $x \to y$ by checking if a cross map $\tilde{\mathbf{y}} \mapsto \tilde{\mathbf{x}}$ exists. To this end, Sugihara et. al. [41] propose to check the predictability of the time series $x(t)$ given $\tilde{\mathbf{y}}(t)$. They use a form of $k$-nearest neighbors regression to produce an estimate $\hat{x}(t)$ of $x(t)$ given $\tilde{\mathbf{y}}(t)$, and then they define a test statistic

$$r_{X \to Y}^{\text{CCM}} = \text{corr}(\hat{x}(t), x(t)), \tag{7}$$

where corr is the Pearson correlation coefficient. To test the reverse causal direction, $x \leftarrow y$, one simply performs Step 2 again with the roles of $x$ and $y$ reversed.

## 2.2 Tangent Space Causal Inference

Takens' theorem and cross maps as a tool for causal inference are rooted in a solid mathematical foundation, but the CCM test does not exploit all of the properties of shadow manifolds. Namely, it does not exploit the fact that the shadow manifolds are copies of the latent manifolds. In practice, there are cases in which CCM learns a cross map that appears to be reasonably predictive, but results in a false positive. Thus, a more robust algorithm would exploit more subtle properties of the hypothesized cross map. We propose TSCI as alternative to Step 2 in the CCM algorithm.

TSCI operates by checking if the ODE on one manifold, $\mathcal{M}_x$, can be mapped to an ODE on another manifold $\mathcal{M}_y$. To understand how this works, we need to reframe our discussion of ODEs in terms of vector fields. Recall that given an ODE of the form,

$$\frac{d\tilde{\mathbf{x}}}{dt} = \mathbf{u}(\tilde{\mathbf{x}}), \tag{8}$$

we may interpret $\mathbf{u}$ to be a velocity vector field on the manifold. When evaluated, $\mathbf{u}(\tilde{\mathbf{x}})$ is a tangent vector of the manifold $\mathcal{M}_x$, existing in the tangent space $T_{\tilde{\mathbf{x}}}\mathcal{M}_x$[4]. From calculus, we know that tangent vectors in $T_{\tilde{\mathbf{x}}}\mathcal{M}_x$ can be mapped to tangent vectors in $T_{F(\tilde{\mathbf{x}})}\mathcal{M}_y$ by the Jacobian matrix $\mathbf{J}_F(\tilde{\mathbf{x}})$ at the point $\tilde{\mathbf{x}}$. By checking if the tangent vectors can be mapped in such a way, TSCI provides an alternative to the CCM causality test. A visual motivation for the TSCI methods is depicted in Fig. 1.

Before we can map vector fields from one manifold to another, we need to check that meaningful vector fields exist on the shadow manifolds. As usual, this is also a corollary of Takens' theorem.

**Corollary 2.1.2** *There exists a vector field $\mathbf{u}$ on $\mathcal{M}_x$ such that the embedding $\Phi_{\mathbf{f},h,\tau}$ in Takens' theorem is a time-invariant mapping. If we define $\gamma(0) = \Phi_{\mathbf{f},h,\tau}(\mathbf{z}_x(0))$, and if we let $\gamma(t)$ to be the flow of a point $\gamma(0)$ under the vector field $\mathbf{u}$, then $\Phi_{\mathbf{f},h,\tau}(\mathbf{z}_x(t)) = \gamma(t)$, for all times $t$.*

---

[4]See Appendix A for definitions of differential geometric quantities such as the tangent space.

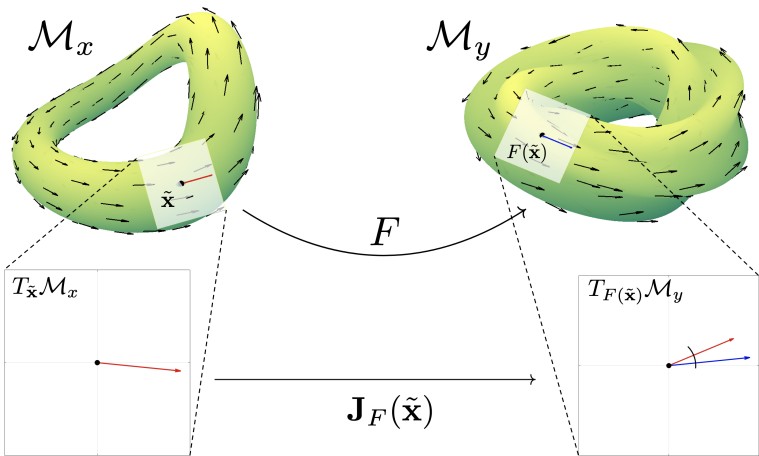

Figure 1: Visual motivation for the TSCI method. Observed time series are related to latent states evolving on some manifolds. Given observed time series $x(t)$ and $y(t)$, we reconstruct latent states $\tilde{\mathbf{x}}(t)$ and $\tilde{\mathbf{y}}(t)$ that reside in manifolds $\mathcal{M}_x$ and $\mathcal{M}_y$, respectively. If $x$ and $y$ are causally coupled, there should exist a function $F$ that maps between these manifolds. Using the Jacobian matrix $\mathbf{J}_F$, we can map the velocity vector field on $\mathcal{M}_x$ to a vector field on $\mathcal{M}_y$, and use the angle between velocity vectors as a measure of their similarity.

The proof of the corollary is provided in the Appendix.

The corollary guarantees that we can learn ODEs that describe the dynamics of $\tilde{\mathbf{x}}$ and $\tilde{\mathbf{y}}$, once we have obtained valid reconstructions of the latent states. Let $\mathbf{u}$ and $\mathbf{v}$ be vector fields such that

$$\frac{d\tilde{\mathbf{x}}}{dt} = \mathbf{u}(\tilde{\mathbf{x}}), \tag{9}$$

$$\frac{d\tilde{\mathbf{y}}}{dt} = \mathbf{v}(\tilde{\mathbf{y}}). \tag{10}$$

Because the cross map $F$ is a mapping between the two shadow manifolds, the Jacobian matrix of the cross map relates the vectors $\mathbf{u}$ and $\mathbf{v}$. We state this formally in the following Lemma.

**Lemma 2.1.1** *Let $\mathcal{M}_x$ and $\mathcal{M}_y$ be manifolds with respective vector fields $\mathbf{u}$ and $\mathbf{v}$ that define their dynamics. If there exists a cross map $F : \mathcal{M}_x \rightarrow \mathcal{M}_y$, then for every point $\tilde{\mathbf{x}} \in \mathcal{M}_x$, we have that $\mathbf{v}(F(\tilde{\mathbf{x}})) = \mathbf{J}_F(\tilde{\mathbf{x}})\mathbf{u}(\tilde{\mathbf{x}})$, where $\mathbf{J}_F(\tilde{\mathbf{x}})$ is the Jacobian matrix of $F$ at $\tilde{\mathbf{x}}$.*

The proof of the lemma is in the Appendix.

While the cross map $F : \mathcal{M}_x \rightarrow \mathcal{M}_y$ is a mapping between points on the manifolds, the Jacobian matrix $\mathbf{J}_F(\tilde{\mathbf{x}})$ induces a mapping between the tangent spaces at $\tilde{\mathbf{x}}$ and $\tilde{\mathbf{y}}$. We show this visually in Fig. 1. Since the velocity of $\tilde{\mathbf{x}}$ can be represented by a tangent vector $\mathbf{u}(\tilde{\mathbf{x}})$, the Jacobian matrix $\mathbf{J}_F$ allows us to map these vectors to tangent vectors in $\mathcal{M}_y$. If there is a cross map, then the vector field $\mathbf{v}$ and the push forward vector field $\mathbf{J}_F\mathbf{u}$ should match exactly. On the other hand, there should be no correlation in the absence of a causal relationship, assuming quality reconstructions and plentiful data. The degree of alignment between $\mathbf{v}$ and $\mathbf{J}_F\mathbf{u}$ can be used as a test statistic for the presence of a causal link. We therefore propose the TSCI test statistic,

$$r_{X \rightarrow Y}^{\text{TSCI}} = \text{corr}(\mathbf{u}, \mathbf{J}_F\mathbf{v}). \tag{11}$$

Because the tangent vectors are centered in their respective tangent planes, we can geometrically interpret the TSCI test statistic as the expected cosine similarity between the vector field $\mathbf{u}$ and the push forward vector field $\mathbf{J}_F\mathbf{v}$,

$$r_{X \rightarrow Y}^{\text{TSCI}} = \mathbb{E}_{\tilde{\mathbf{y}} \sim \mathcal{M}_y} \left( \frac{\mathbf{u}(F(\tilde{\mathbf{y}}))^\top \mathbf{J}_F(\tilde{\mathbf{y}})\mathbf{v}(\tilde{\mathbf{y}})}{||\mathbf{u}(F(\tilde{\mathbf{y}}))|| \, ||\mathbf{J}_F(\tilde{\mathbf{y}})\mathbf{v}(\tilde{\mathbf{y}})||} \right). \tag{12}$$

In practice, the estimation quality of the cross map also depends on the location in the reconstruction space, and so the cosine similarity takes on a distribution of values (Fig. 2), and weak-to-moderate correlation may be empirically observed as an artifact of limited data.

---

**Algorithm 1** TANGENT SPACE CAUSAL INFERENCE

---

**Input:** Embedding/vector field matrices $\mathbf{X}, \mathbf{U} \in \mathbb{R}^{T \times Q_x}$ and $\mathbf{Y}, \mathbf{V} \in \mathbb{R}^{T \times Q_y}$
**Output:** Score $r_{X \to Y}$
  1: Learn a differentiable function $F$ such that $\mathbf{X}_t = F(\mathbf{Y}_t)$ for all $t$
  2: **for** $t = 0, 2, \ldots, T-1$ **do**
  3:      Compute the Jacobian matrix $\mathbf{J}_F(\mathbf{Y}_t) \in \mathbb{R}^{Q_x \times Q_y}$ of the function $F$ at location $\mathbf{Y}_t$
  4:      Push forward the tangent vector by computing $\hat{\mathbf{U}}_t = \mathbf{V}_t \mathbf{J}_F^\top(\mathbf{Y}_t)$
  5: **end for**
  6: Define $r_{X \to Y} = \text{corr}(\hat{\mathbf{U}}, \mathbf{U})$

---

---

**Algorithm 2** TANGENT SPACE CAUSAL INFERENCE WITH K-NEAREST NEIGHBORS

---

**Input:** Embedding/vector field matrices $\mathbf{X}, \mathbf{U} \in \mathbb{R}^{T \times Q_x}$ and $\mathbf{Y}, \mathbf{V} \in \mathbb{R}^{T \times Q_y}$, regression parameter $K > \max(Q_x, Q_y)$
**Output:** Score $r_{X \to Y}$
  1: **for** $t = 0, 2, \ldots, T-1$ **do**
  2:      Find the indices $\tau_1, \ldots, \tau_K$ of the $K$-nearest neighbors of $\mathbf{Y}_t$
  3:      Compute the local displacements in $x$-space: $\Delta\tilde{\mathbf{X}}_k = \mathbf{X}_{\tau_k} - \mathbf{X}_t$
  4:      Compute the local displacements in $y$-space: $\Delta\mathbf{Y}_k = \mathbf{Y}_{\tau_k} - \mathbf{Y}_t$
  5:      Compute the least-squares solution to $\Delta\mathbf{Y}_k \mathbf{J} = \Delta\mathbf{X}_k$
  6:      Compute $\hat{\mathbf{U}}_t = \mathbf{V}_t \mathbf{J}$
  7: **end for**
  8: Define $r_{X \to Y} = \text{corr}(\hat{\mathbf{U}}, \mathbf{U})$

---

Based on these analyses, we propose the TSCI method in Algorithm 1, which learns a cross-map $F$ and returns the alignment of the vector fields $\mathbf{u}$ and $\mathbf{J}_F \mathbf{v}$. CCM is sensitive to the quality of the reconstruction of the latent states because an improperly constructed shadow manifold can sabotage the method, both theoretically and empirically [6]; TSCI similarly requires that the shadow manifold be properly embedded. Because of this, we assume that the embedding vectors $\tilde{\mathbf{x}}$ and $\tilde{\mathbf{y}}$ and the estimates of their vector fields $\mathbf{u}(\tilde{\mathbf{x}})$ and $\mathbf{v}(\tilde{\mathbf{y}})$ at each point are supplied as inputs to the algorithm. A variety of methods could be used to estimate the vector fields, but the simplest approach is to use finite differences,

$$\mathbf{u}(\tilde{\mathbf{x}}) = \frac{d\tilde{\mathbf{x}}}{dt} = \begin{bmatrix} \frac{dx(t)}{dt} \\ \frac{dx(t-\tau)}{dt} \\ \vdots \\ \frac{dx(t-(Q-1)\tau)}{dt} \end{bmatrix} \approx \frac{1}{\Delta t} \begin{bmatrix} x(t+1) - x(t) \\ x(t+1-\tau) - x(t-\tau) \\ \vdots \\ x(t+1-(Q-1)\tau) - x(t-(Q-1)\tau) \end{bmatrix},$$

where $\Delta t$ is the sampling rate of the data. Finite differences are known to be sensitive to noise, so in real scenarios a more careful approach to obtaining the time derivatives is necessary. For a practical implementation, we propose using the central finite-differences, which are second-order accurate [16], or the derivatives interpolated by a Savitsky-Golay filter for noisy data [34].

**On methods to learn the cross map function.** The TSCI algorithm is notably agnostic to the regression approach used for the cross map $F$, with reasonable approaches including multilayer perceptron networks (MLPs), splines, and Gaussian process regression (GPR). Depending on the approach, derivatives can then be estimated from the model, either using automatic differentiation or analytical derivatives. For GPR in particular, analytical derivatives are straightforward to compute [38].

Since regression can sometimes be a computationally complex procedure, in Algorithm 2 we also provide a version of TSCI that is based on the $k$-nearest regression, in analogy to CCM. For clean and dense data, this simple approach can yield accurate results, but it is generally unsuitable for noisy or sparse data. In the latter case, it is preferable to combine the TSCI approach with other methods of denoising time series, or learning of latent dynamical models of the observed time series. In particular, the learning of a latent ODE learning [12] can be combined with the TSCI test to yield accurate causal inference.

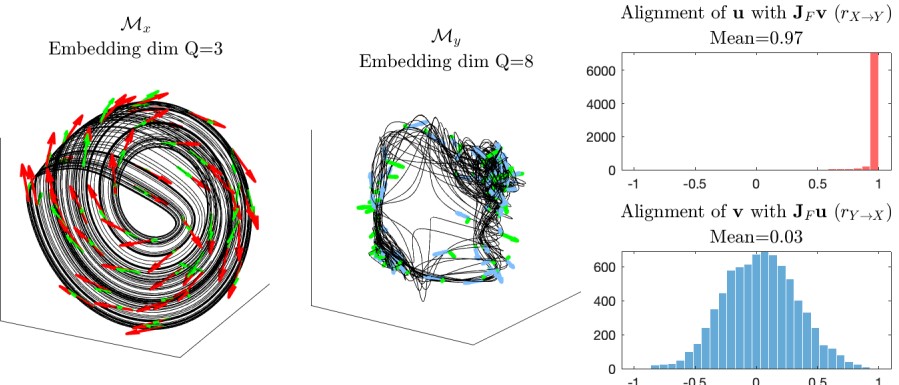

Figure 2: Shadow manifolds $\mathcal{M}_x$ and $\mathcal{M}_y$ from the unidirectionally coupled Rössler-Lorenz system Eq. (13) with $C = 1$, and the corresponding histograms of $\hat{\mathbf{u}}^\top \mathbf{u}$ and $\hat{\mathbf{v}}^\top \mathbf{v}$. The test statistics, $r_{X \to Y}$ and $r_{X \leftarrow Y}$, correspond to the means of these distributions.

**On the use of correlation coefficient.** In relation to CCM, it can be noted that both TSCI and CCM use a correlation coefficient as their test statistic. However, a critical difference between the two methods is how the usage of this statistic is justified. The correlation coefficient used in CCM analysis, called the cross-map skill [41], is used to measure the accuracy of predictions of the cross map. However, since the points being predicted in CCM live on a manifold, measuring correlation in the ambient (extrinsic) space is not well-motivated. Furthermore, estimates of this correlation can be biased by the distribution of observations along the manifold, and as a result, a high correlation can be achieved with a relatively low-accuracy prediction by guessing the general region in which the points lie.

In contrast, the correlation between tangent vectors (or vector fields) is a more geometrically motivated quantity: the correlation is a linear measure of similarity and the tangent vectors belong to a (centered) linear space. Additionally, the shadow manifolds are constructed to be submanifolds of Euclidean space, so correlations computed in extrinsic coordinates will match correlations computed using the intrinsic coordinates of the shadow manifold. By Lemma 2.1.1, the correlations will be identically 1 if a cross map exists.

One alternative to the cosine similarity is the mutual information (MI). Specialized to the current text, the MI $I(\mathbf{u}; \mathbf{J}_F \mathbf{v})$ quantifies the reduction in the uncertainty of $\mathbf{u}$ given the pushforward vector $\mathbf{J}_F \mathbf{v}$ [9]. While the information-theoretic underpinning of the MI is attractive, we have two main reasons to prefer the cosine similarity: First, the MI between two (continuous) distributions can be difficult to interpret. For example, it is not obvious if $I(\mathbf{u}; \mathbf{J}_F \mathbf{v}) = 0.5$ is a strong dependence or not, particularly in the case of continuous distributions. The cosine similarity, on the other hand, is upper bounded by 1, so $\mathrm{corr}(\hat{\mathbf{U}}, \mathbf{U}) = 0.95$ is easily interpreted as having near-perfect reconstruction. Second, accurate estimation of the MI from samples is a notoriously difficult task, particularly in high dimensions, and no single estimator works consistently well [10]. We provide some experiments with the MI, and show the differing performance of different estimators, in Appendix B.3.

## 3 Experiments

We validate the performance of TSCI on two datasets that are popular in the literature. The first arises from a coupled Rössler-Lorenz system, where the ground truth causality is known and the causal influence can be smoothly modulated. The second example was proposed in [12] and uses sporadic time series from coupled double pendulums, and illustrates the applicability of TSCI to extensions of CCM. Code implementing TSCI is available at `https://github.com/KurtButler/tangentspaces`. All comparisons to CCM use the `skccm` module in Python[5]. Experiments were run on a 6-Core Intel Core i5 and NVIDIA Titan RTX.

---

[5]`https://github.com/nickc1/skccm`, MIT License

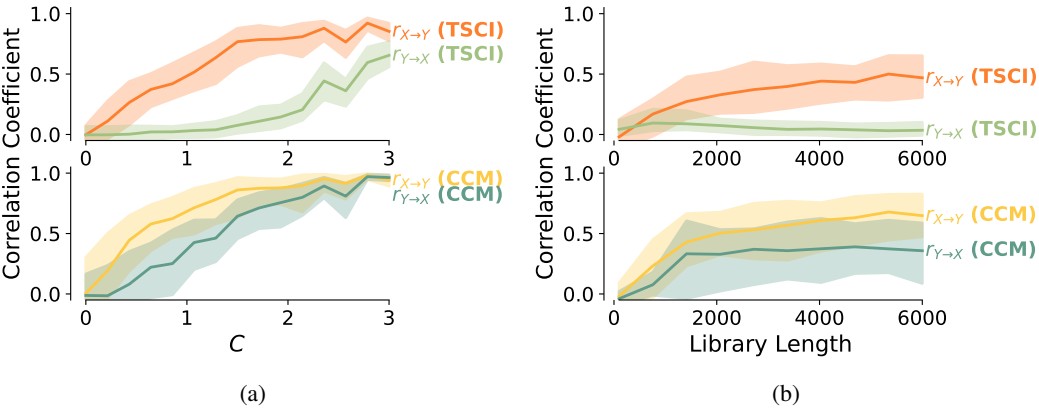

Figure 3: Comparison of TSCI with CCM for the Rössler-Lorenz system (true causation $X \to Y$). The plotted lines show the median test statistic over 100 trials for both CCM and TSCI, and the shaded region indicates the 5th and 95th percentiles when (a) $C$ is varied from 0 (no coupling) to 3 (approximate general synchrony), and (b) $C$ is fixed to $1.0$ and the library length is varied.

### 3.1 Unidirectionally-Coupled Rössler-Lorenz System

A common toy system used for studying coupled dynamic systems is a unidirectionally-coupled Rössler-Lorenz system [28], which we define in Eq. (13). In this system, the first three coordinates $(z_1, z_2, z_3)$ describe a Rössler system, and they are unidirectionally coupled with a Lorenz-type system $(z_4, z_5, z_6)$. The strength of the coupling is controlled by the parameter $C$. When $C = 0$, the two systems are disconnected, but for $C > 0$ there is a causal influence from $z_1, z_2, z_3$ to $z_4, z_5, z_6$.

$$\frac{d\mathbf{z}}{dt} = \frac{d}{dt} \begin{bmatrix} z_1 \\ z_2 \\ z_3 \\ z_4 \\ z_5 \\ z_6 \end{bmatrix} = \begin{bmatrix} -6(z_{2,t} + z_{3,t}) \\ 6(z_{1,t}) + 0.2z_{2,t} \\ 6\left(0.2 + z_{3,t}(z_{1,t} - 5.7)\right) \\ 10(z_{5,t} - z_{4,t}) \\ 28z_{4,t} - z_{5,t} - z_{4,t}z_{6,t} + Cz_{2,t}^2 \\ z_{4,t}z_{5,t} - 8z_{6,t}/3 \end{bmatrix} \tag{13}$$

In Fig. 2, we visualize the TSCI method for $x = z_2$ and $y = z_4$. First, we show the shadow manifolds $\mathcal{M}_x$ and $\mathcal{M}_y$ with a set of tangent vectors on each manifold. The delay embedding dimension parameters, $Q_x$ and $Q_y$, were selected using the false-nearest neighbors algorithm with a tolerance of 0.005 [19]. Time lags for the embeddings, $\tau_x$ and $\tau_y$, were selected by picking the minimal delay such that the autocorrelation function drops below a threshold [21]. Additionally, we show the histograms of $\cos(\theta)$, where $\theta$ is the angle between the tangent vectors at each point. For $X \to Y$ direction, which is the true causal direction, the distribution is concentrated near 1. For the $Y \leftarrow X$ direction, the distribution of tangent vectors is centered on 0. The test statistics, $r_{X \to Y}$ and $r_{Y \to X}$, correspond to the means of each distribution, and visibly correspond to the correct causality.

We show the effects of varying the coupling strength $C$ in Fig. 3a, where TSCI clearly shows better separation across varying $C$. We see the effect of increasing the library length (i.e., the size of the training set in nearest neighbors) for $C = 1.0$ in Fig. 3b. Here, $r_{X \to Y}$ increases at similar rates for TSCI and CCM, suggesting similar data efficiency of the two methods.

### 3.2 Double Pendulum System

To illustrate the generality of TSCI within CCM-like frameworks, we applied the TSCI methodology to the latent CCM framework, where CCM is applied to a state-space reconstruction obtained via neural ODEs [12]. One reason to favor this approach is when the observed time series are irregularly (i.e., with a non-uniform sampling rate) or sporadically (i.e., any given observation only measures a subset of states) sampled. Notable for TSCI is the fact that ground-truth derivatives are available, as they are directly learned in the neural ODE reconstruction.

Table 1: Double Pendulum Experiments. Entries denote the mean $\pm$ one standard deviation across 5 folds. Bolded directions indicate ground truth causality.

| Direction | $r_{X_1 \to X_2}$ | | |
|---|---|---|---|
| | Latent CCM | Latent CCM (MLP) | Latent TSCI (MLP) |
| $X \to Y$ | $0.011 \pm 0.009$ | $0.015 \pm 0.009$ | $0.021 \pm 0.010$ |
| $X \to Z$ | $0.044 \pm 0.008$ | $0.055 \pm 0.011$ | $0.077 \pm 0.013$ |
| $Y \to X$ | $-0.003 \pm 0.008$ | $-0.005 \pm 0.009$ | $-0.010 \pm 0.004$ |
| $Y \to Z$ | $0.003 \pm 0.006$ | $-0.002 \pm 0.004$ | $-0.008 \pm 0.013$ |
| $\mathbf{Z \to X}$ | $0.737 \pm 0.019$ | $0.747 \pm 0.015$ | $0.915 \pm 0.006$ |
| $\mathbf{Z \to Y}$ | $0.475 \pm 0.043$ | $0.578 \pm 0.030$ | $0.612 \pm 0.053$ |

We replicate an experiment in the latent CCM paper where a network of unidirectionally coupled double pendulums are simulated, and observations sampled irregularly and sporadically. The authors' source code[6] was used to generate data and apply latent CCM. For more information on the data generation, see Appendix A of [12].

For TSCI, we learn a cross-map using an MLP between the reconstructed state spaces. To avoid tuning learning rates for every network, the parameter-free COCOB optimizer [26][7] and networks were trained for 50 epochs. Results can be found in Table 1.

When compared to latent CCM, TSCI provided larger correlation coefficients for the true positive cases, with similarly small coefficients for the true negative cases. The use of MLPs for learning the cross map partially but not fully explains the difference in performance.

### 3.3 Additional Experiments

Several additional experiments appear in Appendix B. We briefly describe them here and provide some commentary on their results.

**Varying the embedding dimension.** In Appendix B.1, we artificially lower quality embeddings by varying the embedding dimension. We find that the performance of TSCI and CCM similarly degrade when using an embedding dimension that is very small, and that there is little harm to a larger embedding dimension. In either case, the hyperparameters from a false-nearest neighbors test perform well.

**Corrupting signals.** In Appendix B.2, we lower the embedding quality in two different ways: (1) by injecting additive noise to all observations, and (2) by adding a sine wave to observations. We observe that TSCI and CCM similarly degrade when the signal-to-noise ratio is altered. However, because of the larger separation in TSCI, the correct causal relation can be determined at much lower signal-to-noise ratios.

**Using mutual information.** In Appendix B.3, we experiment with using mutual information instead of cosine similarity. We find that conclusions made using MI are generally similar to those from cosine similarity, but with less interpretability and significant difficulties in estimation.

**Comparisons to other causal discovery methods.** As mentioned in the introduction, CCM (and hence TSCI) are specifically designed to address failures of Granger causality. In Appendix B.4, we empirically verify the limitations of Granger causality on our Rössler-Lorenz toy system. We additionally test several other causal discovery methods and show their limitations in our setting.

## 4 Advantages and Limitations of TSCI

**Scalability.** The TSCI approach, using the $K$-nearest neighbors algorithm, retains the scalability and lightweight implementation that CCM enjoys. The only additional computational complexity arises

---

[6]`https://github.com/edebrouwer/latentCCM`, MIT License

[7]`https://github.com/bremen79/parameterfree`, MIT License

from solving a local linear system of equations, and from the estimation of derivatives. Since the linear systems have a fixed size $K$, and since derivative estimation can be done using a linear filter, the additional cost is minimal. The general version of TSCI will scale depending on the choice of the regressor used to learn the cross map, and on other design decisions made to improve the efficacy of the method. However, this is not unlike the situation in which CCM is augmented with other customization options that potentially slow down the method.

**Model agnosticism.** Because a TSCI test can be formulated for any differentiable regression model used to learn the cross map, the approach is highly flexible and model agnostic. Additionally, because there are many potential ways in which reconstruction of latent states and learning of the velocity vector fields can be improved, the TSCI method can be incorporated into a wide variety of inference frameworks.

**Quality of reconstructed states.** In both CCM and TSCI, there are a few assumptions which warrant some justification before use, since their violation may yield to misapplication of cross map methods. The first issue is that while Takens' theorem implies that embeddings are plentiful, not all embeddings are equal in quality or useful. Shadow manifolds that are sparsely or incompletely sampled, time series with trends, or otherwise data which do not accurately capture the latent manifold can lead to dubious results using CCM [6]. While TSCI is less likely to produce a false positive in these cases, the assumption that a latent manifold is well-represented by a given embedding is nontrivial and critical to ensuring trustworthy performance of the method.

**General synchrony.** General synchrony is a problem that plagues all cross map-based methods [41]. The issue is that when the causal strength of the relationship $x \to y$ is very strong, the influence of $\mathbf{z}_x$ dominates the dynamics of $\mathbf{z}_y$, and $\mathbf{z}_y$ cannot exhibit its own independent behavior. As a result, $\mathcal{M}_y$ will look similar to $\mathcal{M}_x$, and the cross map will become an invertible function. As a result, a strong unidirectional relationship is detected as a bidirectional causal relationship. In Fig. 3, the Rössler-Lorenz system enters general synchrony near $C = 3$ [6]. The TSCI method appears more resistant to the effects of general synchrony than CCM, but it is a topological fact that as $C$ grows, synchrony becomes inevitable.

## 5   Conclusion

In this paper, we presented the TSCI method for detecting causation in dynamical systems. By considering how tangent vectors map from one manifold to another, we may achieve more robust detection of causal relationships in dynamical systems than with standard CCM. Key advantages of TSCI include that we may use it in many systems in which CCM would be applied, but the method is far less prone to false positives and spurious causation. We presented both a general form of the algorithm as well as a $k$-nearest neighbor version inspired by the original CCM algorithm. Because TSCI requires us to estimate latent states and their time derivatives, there us much room for the TSCI method to be further developed in future work.

## Acknowledgments and Disclosure of Funding

We would like to thank the anonymous reviewers for their suggestions which improved the content and presentation of this work. This work was supported by the National Science Foundation under Award 2212506.

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

# A   Important Facts About Manifolds

In this appendix, we review some important background about manifolds. Much more complete references include the authoritative [22] (especially chapters 1, 2, 3, and 8), or the more accessible [24].

We say that a function $F$ is *smooth* if it has derivatives of all orders. A *diffeomorphism* is a function $F$ which is smooth, invertible, and $F^{-1}$ is also smooth. We say that two subsets $M$ and $N$ of $\mathbb{R}^n$ are diffeomorphic if there exists a *diffeomorphism* such that $F(M) = N$.

A subset $M$ of $\mathbb{R}^n$ is called a $d$-**dimensional manifold** if for every point $x$ in $M$, there exists an open set $U_x$ such that

1. There is a diffeomorphism $F_x : \mathbb{R}^n \to U_x$.

2. The image of the set $\mathbb{H}_d = \{y \in \mathbb{R}^n : y_{d+1} = \cdots = y_n = 0\}$ under $F_x$ is given by

$$F_x(\mathbb{H}_d) = U_x \cap M.$$

Without a loss of generality, we can assume that $F_x(0) = x$ in this definition. This allows us to easily define tangent vectors. A vector $v \in \mathbb{R}^n$ is called a **tangent vector** of $M$ at the point $x$ if there exists a path $\gamma(t)$ in $\mathbb{H}_d$ such that

$$\frac{d}{dt}\left(F_x(\gamma(t))\right|_{t=0} = v,$$

where $F_x$ is the diffeomorphism in the above definition. The **tangent space** $T_x M$ is defined to be the set of all tangent vectors of $M$ at $x$. Note that from these definitions, it is clear that $\mathbb{R}^n$ is a manifold and $T_x \mathbb{R}^d = \mathbb{R}^d$.

In calculus, we define a vector field $\mathbf{u}$ on $\mathbb{R}^n$ to be a function $\mathbf{u} : \mathbb{R}^n \to \mathbb{R}^n$. We generalize this to manifolds in Euclidean space by defining a **vector field $\mathbf{u}$ on** $M$ to be a mapping $\mathbf{u} : M \to \mathbb{R}^n$ such that $\mathbf{u}(x) \in T_x M$ at every point $x$.

An *embedding* of a manifold $M \subset \mathbb{R}^n$ is a mapping $F : \mathbb{R}^n \to \mathbb{R}^m$ such that $F$ restricted to $M$ is a diffeomorphism onto its image. If $M$ is a manifold of dimension $d$ and $F$ is an embedding, then $m$ is necessarily greater than or equal to $d$. In general, it is an interesting question to ask how much larger $m$ must be so that embeddings are easy to find. The Whitney embedding theorem says that it is a generic property that $F : \mathbb{R}^n \to \mathbb{R}^m$ when $m \geq 2d + 1$. Takens' theorem refines the Whitney theorem by asserting that such an embedding can be obtained from a scalar observation function, by time lagging the observations sufficiently many times.

# B   Additional Experiments

In this appendix, we include experiments and results omitted from the main text.

## B.1   Varying Embedding Dimension

One of the key parameters in both CCM an TSCI is the embedding dimension parameter $Q$ used during reconstruction of the shadow manifold. Takens' theorem suggests that once $Q$ reaches a certain value, the manifold is embedded and the improvement of the results should saturate in principle. Of course, in real life there is a curse of dimensionality associated with inference, so it is interesting to check how well the results behave as the embedding dimension is chosen to be higher than necessary. We consider the effect of varying the parameter $Q$ using data from the Rössler-Lorenz system in Fig. 4. Performance of both methods improve as the dimension of the putative effect (if we are testing $x \to y$, then this is $Q_y$) is increased, until saturating after the manifolds are fully embedded.

## B.2   Corrupted Signals

Generally, there are many non-trivial ways in which the state reconstruction can be poor, e.g., presence of periodic trends [8] and underexploration of the shadow manifold [6]. We consider two scenarios that deteriorate the state reconstruction quality on the Rössler-Lorenz system: additive noise, and a periodic trend.

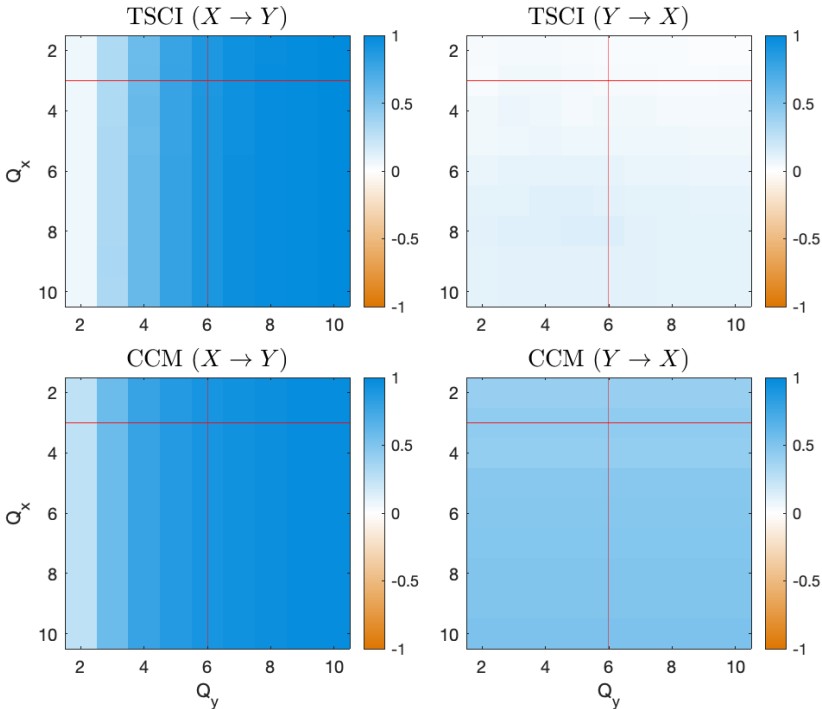

Figure 4: Comparison of TSCI and CCM for the unidirectionally coupled Rössler-Lorenz system Eq. (13) with $C = 1$. Here, $x(t) = z_2(t)$ and $y(t) = z_4(t)$, and the true causality is $x \to y$. The red lines indicate the values of $Q_x$ and $Q_y$ selected by false-nearest neighbors with a tolerance of 0.01.

Results for additive noise are presented in Fig. 5. We fixed $C = 1.0$ and varied the signal-to-noise ratio (SNR) from 0 dB to 60 dB by adding Gaussian noise to the observed signals. When estimating derivatives, a Savitzky-Golay filter [34] was used with a window length of 5 and order 2.

Overall, we find the performance relative to SNR to be similar between CCM and TSCI, suggesting that inference of the vector field from noisy data is not a limiting factor in the presence of additive noise.

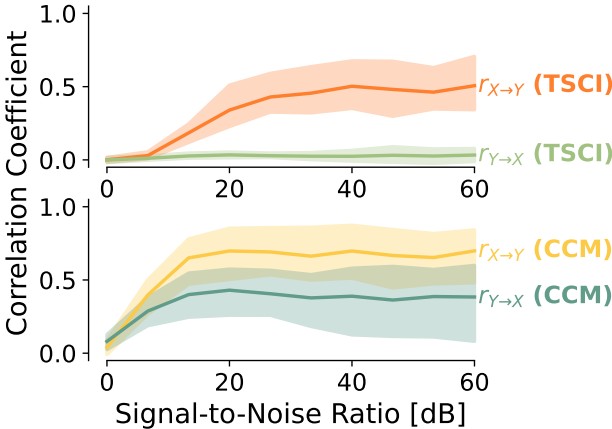

Figure 5: Comparison of TSCI with CCM for the Rössler-Lorenz system (true causation $X \to Y$) with additive noise. The plotted lines show the median test statistic over 100 trials for both CCM and TSCI, and the shaded region indicates the 5th and 95th percentiles when (a) $C$ is fixed to 1.0 and the signal-to-noise ratio is varied.

An alternative way to corrupt signals is with a deterministic periodic trend; we used a sine wave with a period of $2\pi$, and varied its power with respect to the true signal. Results are presented in Fig. 6. Our takeaways are relatively similar to the additive noise experiment: TSCI seems to degrade similarly to CCM, but its increased separation between cause and effect makes this degradation in performance less problematic. Notably, TSCI seems significantly more robust to false claims of strong causation when the relative power of the confounder is large.

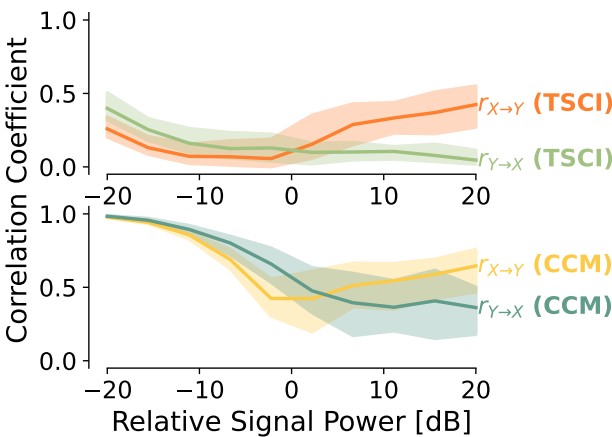

Figure 6: Results of the Rössler-Lorenz system with $C = 1$ corrupted by an additive sine signal of varying amplitude. The amplitude of the signal, relative to the sine wave, is plotted in decibels on the horizontal axis.

## B.3 Using the Mutual Information for Scores

One interesting alternative to the cosine similarity score is the mutual information (MI), which provides a non-linear quantification of how related $\mathbf{u}$ and $\mathbf{J}_F\mathbf{v}$ are. One immediate concern with using MI is that estimates from finite samples are difficult; we use the classical estimator of Kraskov, Stögbauer, and Grassberger [20], which has shown relevance even in the era of deep learning [10].

Results on the Rössler-Lorenz test system while varying the coupling parameter $C$ are presented in Fig. 7. We find that MI typically shows less separation than TSCI, which brings into question its viability when the ground truth is unknown. Another issue is in the interpretation of MI estimates: it is not entirely clear if an MI estimate of, for example, 1 nat indicates strong influence, and MI estimates are not easily normalized in the continuous setting.

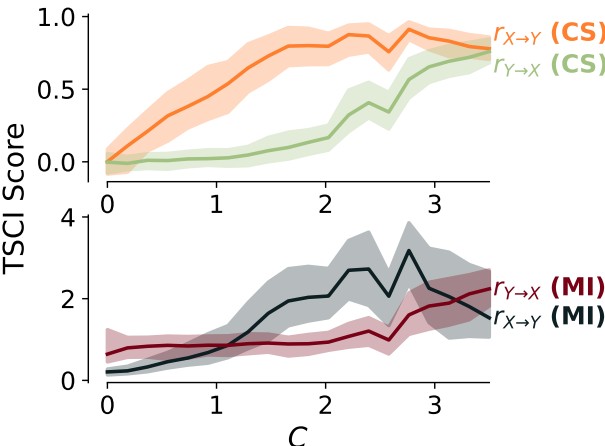

Figure 7: Results of the Rössler-Lorenz system using the cosine similarity and mutual information while varying $C$. Compare to Figure 2a in the main manuscript.

## B.4  Comparisons to Other Causal Discovery Methods

As argued in the main manuscript, cross-map-based methods such as TSCI and CCM approach causal discovery in a rather particular setting, where more mainstream methods do not apply. We empirically verify these claims on the Rössler-Lorenz test system, using a variety of values of $C$. We compare against Granger causality, as well as various bivariate causal discovery methods.

We first compare against Granger causality in Table 2. As expected from the theory, Granger causality incorrectly infers bidirectional causality for all $C > 0$.

We also compare against various methods from the bivariate or pairwise causal discovery literature. These methods typically assume the existence of a map $\mathbf{y} = f(\mathbf{x})$ which is either deterministic or subject to additive noise, which does not exist in all but the most trivial dynamical systems. This includes RECI [4], IGCI [11], and ANM [17]. Results are presented in Tables 3 and 4.

We find that IGCI consistently fails to detect a causal edge, with weak coefficients. RECI consistently chooses the direction $X \to Y$, but does so even when $C = 0$, which suggests that results are not actually detecting causality, but rather some other anomaly of the data. To this end, the variance ratios as described by Blobaum et al. [4] are typically large, indicating little confidence. Finally, for all $C > 0$, ANM detects bidirectional causality.

Comparisons with Granger causality use the implementation of `statsmodels`[8] [37]. All comparisons to RECI and IGCI used the implementations of `cdt`[9] [18], with a minor bug-fix for RECI. Comparisons with ANM used the implementation of `causal-learn`[10] [46].

Table 2: Granger Causal Results for Rössler-Lorenz System. Entries denote the median, 5th, and 95th percentile p-values across 50 trials. All p-values are from the F-test implemented in `statstools`.

| $C$ | p-value $X \to Y$ | p-value $Y \to X$ |
|-----|-------------------|-------------------|
| 0.0 | $0.003_{0.000}^{0.345}$ | $0.168_{0.000}^{0.896}$ |
| 0.75 | $0.000_{0.000}^{0.000}$ | $0.000_{0.000}^{0.000}$ |
| 1.5 | $0.000_{0.000}^{0.000}$ | $0.000_{0.000}^{0.000}$ |
| 2.25 | $0.000_{0.000}^{0.000}$ | $0.000_{0.000}^{0.000}$ |
| 3.0 | $0.000_{0.000}^{0.000}$ | $0.000_{0.000}^{0.000}$ |

Table 3: Bivariate causal discovery results for Rössler-Lorenz System. For all models, a negative score indicates $X \to Y$ and a positive score indicates $Y \to X$. All scores are based on the implementations of `cdt`. Entries denote the median, min, and max over 10 trials.

| $C$ | RECI | IGCI |
|-----|------|------|
| 0.0 | $-0.021_{-0.027}^{-0.017}$ | $0.035_{0.018}^{0.074}$ |
| 0.75 | $-0.025_{-0.048}^{-0.019}$ | $0.034_{-0.017}^{0.048}$ |
| 1.5 | $-0.028_{-0.039}^{-0.025}$ | $-0.016_{-0.078}^{0.031}$ |
| 2.25 | $-0.029_{-0.036}^{-0.027}$ | $0.040_{0.004}^{0.073}$ |
| 3.0 | $-0.036_{-0.041}^{-0.017}$ | $0.030_{0.011}^{0.042}$ |

## C  Proofs

In this section, we provide proofs for results mentioned in the main text.

**Proof of Theorem 2.1**  There are many different proofs and versions of Takens' theorem. This version is proved by Sauer, Yorke and Casdagli [33], and is beyond the scope of this appendix. Other noteworthy variations of this theorem are proved by Stark [39] and Stark et al. [40].  □

---

[8] `https://github.com/statsmodels/statsmodels/`, BSD-3-Clause License

[9] `https://github.com/FenTechSolutions/CausalDiscoveryToolbox`, MIT License

[10] `https://github.com/py-why/causal-learn`, MIT License

Table 4: ANM causal discovery results for Rössler-Lorenz System. Entries denote the median, min, and max $p$-value over 10 trials. All scores are based on the implementation in `causal-learn`.

| $C$ | $X \to Y$ | $Y \to X$ |
|---|---|---|
| 0.0 | $0.507^{0.777}_{0.096}$ | $0.450^{0.915}_{0.003}$ |
| 0.75 | $0.000^{0.121}_{0.000}$ | $0.000^{0.049}_{0.000}$ |
| 1.5 | $0.000^{0.000}_{0.000}$ | $0.000^{0.000}_{0.000}$ |
| 2.25 | $0.000^{0.000}_{0.000}$ | $0.000^{0.000}_{0.000}$ |
| 3.0 | $0.000^{0.000}_{0.000}$ | $0.000^{0.000}_{0.000}$ |

**Proof of Corollary 2.1.1**    Let us assume that we have a generic system, where $(\mathbf{z}_x, \mathbf{z}_y) \in M$ for some manifold $M$, and $x$ is a function $\mathbf{z}_x$ and $y$ is a function of $\mathbf{z}_y$.

If $x \to y$ and $x \leftarrow y$ both hold, then $\mathcal{M}_x$ and $\mathcal{M}_y$ are both diffeomorphic to $M$, and thus, diffeomorphic to each other. The existence of a cross map is then immediate.

If $x \to y$ but not vice-versa, then the dynamics of $\mathbf{z}_x$ are autonomous, meaning that the behavior of $\mathbf{z}_x$ depends only upon itself. As a result, the projection map $\pi_x(\mathbf{z}_x, \mathbf{z}_y) = \mathbf{z}_x$ can be applied directly to $M$ to yield a new manifold $M_x$ on which $\mathbf{z}_x$ is an autonomous dynamical system. Since $x$ observes this system $\mathbf{z}_x$, $\mathcal{M}_x$ will be diffeomorphic to $M_x$, since $\mathbf{z}_x$ only contains information about $\mathbf{z}_x$. Let $\Phi_x : M_x \to \mathcal{M}_x$ and $\Phi_y : M \to \mathcal{M}_y$ denote the diffeomorphisms implied by Takens' theorem. We can construct the cross map then as

$$F(\tilde{\mathbf{y}}) = \Phi_x(\pi_x(\Phi_y^{-1}(\tilde{\mathbf{y}}))).$$

$\square$

**Proof of Corollary 2.1.2**    Since the shadow manifold $\mathcal{M}$ produced by Takens' theorem is a smooth embedding of the original manifold $M$, there exists a diffeomorphism $\Phi : M \to \mathcal{M}$. By Proposition 8.19 of Lee [22, p. 183], for any given vector field $\mathbf{u}$ defined on $M$, there exists a unique vector field $\mathbf{v}$ on $\mathcal{M}$ that is the push forward of $\mathbf{u}$ under the map $\Phi$. Because the dynamics on the manifold are derived from these vector fields, the two systems are equivalent as dynamical systems. $\square$

**Proof of Lemma 2.1.1**    Suppose that $F : \mathcal{M}_x \to \mathcal{M}_y$ is a cross map. Fix any point $\tilde{\mathbf{x}}_0 \in \mathcal{M}_x$ and let $\tilde{\mathbf{y}}_0 = F(\tilde{\mathbf{x}}_0)$. If we consider flows forwards and backwards with respect to these vector fields, then there exists some small $\varepsilon > 0$ such that we have curves $\tilde{\mathbf{x}}_t$ along $\mathcal{M}_x$ and $\tilde{\mathbf{y}}_t$ along $\mathcal{M}_y$ such that $\tilde{\mathbf{y}}_t = F(\tilde{\mathbf{x}}_t)$ for each $t$ in the interval $(-\varepsilon, \varepsilon)$. By the chain rule of calculus, we have that

$$\frac{d\tilde{\mathbf{y}}_0}{dt} = \mathbf{J}_F(\tilde{\mathbf{x}}_0)\frac{d\tilde{\mathbf{x}}_0}{dt},$$

where $\mathbf{J}_F(\tilde{\mathbf{x}})$ is the Jacobian matrix of the function $F$ at the point $\tilde{\mathbf{x}}$. Since the dynamics of these spaces are given by vector fields, we can replace the time derivatives in the equation with their vector fields:

$$\mathbf{v}(\tilde{\mathbf{y}}_0) = \mathbf{J}_F(\tilde{\mathbf{x}}_0)\mathbf{u}(\tilde{\mathbf{x}}_0).$$

Since the point $\tilde{\mathbf{x}}_0$ that we chose was arbitrary, the statement holds for every point in $\mathcal{M}_x$. $\square$

