# OpenReview forum: "Tangent Space Causal Inference: Leveraging Vector Fields for Causal Discovery in Dynamical Systems"
_NeurIPS.cc/2024/Conference — NeurIPS 2024 poster_

### Official Review · Reviewer_rcKb · 2024-06-28

**Soundness:** 3
**Presentation:** 4
**Contribution:** 3
**Rating:** 6
**Confidence:** 4

**Summary:**

This paper presents a novel algorithm to perform causal inference in time series data based on the idea of convergent cross mappings (CCMs). Similar to Granger causality, the CCM paradigm infers causation by testing whether there exists a map (i.e., a predictor) from the reconstructed (unobserved) state trajectory of the effect variable to the reconstructed state trajectory of a cause variable. However, TSCI proposes to compare the (correlations of the) vector fields of the reconstructed dynamical systems rather than the reconstructed trajectories, as done by classical CCM. The approach is validated in two, low-dimensional synthetic examples.

**Strengths:**

Overall, the paper proposes a novel algorithmic idea that is well-motivated in the context of existing CCM literature. References to related work appear complete, but I am not an expert on Granger/CCM causal inference specifically. The paper is very well written, which makes it easy to understand the concept without having explicit background in this field of causal inference, except for certain additional background that could improve understanding further (see below).

**Weaknesses:**

The explanation of why the correlation coefficient over tangent vectors is better than over trajectories (in CCM) is shaky and needs to be made more precise (last paragraph of Section 2). Specifically, it would be great if the work could expand on the argument and reasoning behind: “since the points being predicted in CCM live on a manifold, measuring correlation in the ambient (extrnistic) space is not well-motivated (l. 205-206)”. Why is it not well-motivated? Given this, the work states “the tangent planes [of TSCI are] geometrically motivated” (l. 209-210). Please elaborate more here as well. In my eyes, this paragraph lies at the heart of the differences/(dis)advantages of CCM vs TSCI, so it would be great if this discussion is expanded and made more explicit.

The experimental evaluation of the work can be improved. First, as acknowledged, e.g. in lines 180-183, TSCI requires accurate embeddings/reconstructions of the state space trajectories to work well. However, none of the experiments study how the performance of TSCI and classical CCM behave/degrade as reconstruction quality decreases, which seems like a fundamental component of applying either method in practical applications.
In addition, in none of the experiments, CCM or latent CCM (i.e., existing works) fail to identify or misidentify the true causal effects. Thus, all baselines and TSCI achieve perfect accuracy, which suggests that the experiments may be too easy? The example systems here are very low dimensional, and the library length very long (e.g. Fig 3b). How does the performance compare in larger dimensionalities, where certain edges may be misclassified? The experiments do not distinguish which method is the better one.

Algorithms 1 and 2 are stated for 2-variable dynamical systems (with variables X and Y). Is the extension to multivariate systems simply done by applying TSCI to all pairs of variables in the system (e.g. in the experiments of Section 4.2)? Is there a curse of dimensionality involved when comparing vector fields, because we have to sample tangent vector points on the manifold? The work states that “[CCM] estimates [are] biased by the distribution of observations along the manifold” (l. 206-)”. If TSCI does not have this bias, it must sample from the whole manifold, or am I getting this wrong?

**Questions:**

Some of these questions do not concern TSCI specifically, but CCM in general, and it would improve the quality of the paper if they are explained more explicitly throughout the paper. I am not an expert specifically on Granger causality and CCM, so it would be very helpful to the reader to highlight key differences more clearly.

-	It would be great if Section 2.1. provided some basic mathematical definition and/or background on manifolds, to the degree that is needed to understand the paper. The notion of manifolds appears repeatedly, so some basic intuitions would help a lot (e.g., how do you define a manifold defined? Is a compact subspace of Euclidean space also covered by Taken’s theorem? How does this concept apply, e.g., to the experiments done in Section 3?)

-	Why is it the case that “the angle […] is the same regardless if it is computed from intrinsic or extrinsic coordinates”? (l. 212)

-	What is the fundamental difference, especially conceptually and algorithmically, between CCM and Granger causality? It seems that both ideas are based on learning predictors from one time series to the other. Is the main difference the direction of the predictor (anticausal for CCM, causal for Granger) or the fact that CCM-based methods work with reconstructed trajectories?

-	In general, what are the identifiability conditions or theoretical guarantees of CCM? Can we always infer all causal dependencies in the infinite sample limit for arbitrarily large dynamical systems? From the causal discovery/causal structure learning perspective, this seems like a “free lunch” situation and too good to be true. What assumptions on the class of dynamical systems are needed to guarantee identifying the causal dependencies?

-	Could you give more explanation for why this is the case in the introduction? “… for many systems, namely dynamical systems, the separability condition [of Granger causality] is violated” (l. 32-33)

**Limitations:**

The paper is doing a good job at discussing possible disadvantages of the methods in Section 4. For other limitations, see weaknesses/questions above.

---

> ### Author Rebuttal · Authors · 2024-08-07
>
> We thank the reviewer for their time and consideration. We now address the weaknesses and questions mentioned above.
>
> # Weaknesses
> > The explanation of why the correlation coefficient over tangent vectors is better than over trajectories (in CCM) is shaky
>
> One major reason that the use of correlation in TSCI is more principled over CCM is because it appears naturally when checking the degree of alignment between vector fields, measured by the angle. This correlation is *not* the same as Pearson correlation, because the tangent vectors are not i.i.d. Euclidean vectors and they technically exist in separate tangent spaces, but from the algorithmic perspective, correlation is the operation that results from the math.
> In contrast, using the correlation between points on the manifold is not well-motivated because it does not clearly arise from any geometric principle related to the problem at hand.
>
>
> > TSCI requires accurate embeddings/reconstructions of the state space trajectories to work well. However, none of the experiments study how the performance of TSCI and classical CCM behave/degrade as reconstruction quality decreases
>
> We refer to the general response to reviewers above for comments on how  CCM/TSCI is sensitive to the reconstruction quality. We note that two of the supplemental experiments in the appendix A.1 and A.2) address the degradation of performance due to under-embedding the manifolds, and due to the injection of additive noise. We have also added an additional experiment including a sinusoidal confounder to the general response.
>
> > In addition, in none of the experiments, CCM or latent CCM (i.e., existing works) fail to identify or misidentify the true causal effects. ... The experiments do not distinguish which method is the better one.
>
> For each value of C in Figure 3, a different system was tested. In every case $C > 0$, the true causality is $X \rightarrow Y$. For moderate values of C, CCM concludes that $X \leftrightarrow Y$, which is an incorrect inference. In contrast, TSCI demonstrates much better separation in these experiments.
>
>
> > Algorithms 1 and 2 are stated for 2-variable dynamical systems (with variables X and Y). Is the extension to multivariate systems simply done by applying TSCI to all pairs of variables in the system?
>
> The application of TSCI to multivariate systems proceeds largely in the same way as CCM. The general scheme is to perform this analysis pairwise across the entire network. There are likely more efficient ways to perform this inference, and we generally do not recommend TSCI (or CCM) for large networks, but this is not the focus of the current paper.
>
> > Is there a curse of dimensionality involved when comparing vector fields, because we have to sample tangent vector points on the manifold?
>
> Because the tangent vectors can be estimated from the time derivatives of the original scalar time series, we do not anticipate a strong dependence on the dimension of the manifold. In particular, we expect in general that the tangent vectors belong to a lower dimensional subspace inherited from $\mathcal{M}_X$. See also the experiment in Appendix A.1.
>
> > The work states that “CCM estimates are biased by the distribution of observations along the manifold”. If TSCI does not have this bias, it must sample from the whole manifold, or am I getting this wrong?
>
> The TSCI test statistic uses samples from the whole manifold, as illustrated in Eq. (12). In particular, we take the expected value of the cosine similarity across all test samples.
>
>
> # Questions
> > It would be great if Section 2.1. provided some basic mathematical definition and/or background on manifolds, to the degree that is needed to understand the paper.
>
> Thank you for this suggestion. To make the manuscript easier to read, we are adding some background material on manifolds and Takens' theorem to the appendix. The concepts apply to the systems in Section 3 because they are all generated by systems governed by deterministic differential equations.
>
>
> >  Why is it the case that “the angle is the same regardless if it is computed from intrinsic or extrinsic coordinates”?
>
> The angle between tangent vectors is the same when computed using the intrinsic definition or the extrinsic definition. In technical language, the tangent space $T_x(M)$ at a point x along a manifold $M$ is a vector subspace of the tangent space $T_x(\mathbb{R}^n)$ \[1, p.80\]. Calculation of angles in the space $T_x(M)$ corresponds to an intrinsic coordinate definition, and calculation in $T_x(\mathbb{R}^n)$ corresponds to an extrinsic definition. The angle between two tangent vectors $u$ and $v$ in $T_x(M)$ is the same as if it were computed in $T_x(\mathbb{R}^n)$, due to the subspace property.
>
>
> > What is the fundamental difference, especially conceptually and algorithmically, between CCM and Granger causality?
>
> Please see our general response to reviewers above for discussion about CCM and Granger causality.
>
> > what are the identifiability conditions or theoretical guarantees of CCM? Can we always infer all causal dependencies in the infinite sample limit for arbitrarily large dynamical systems?
>
> In the large sample limit, the shadow manifold represents the reconstructed latent states perfectly. A key assumption here is that the system is generic, as to avoid situations where symmetries of the model inhibit our inference. However, even given a deterministic dynamical system with attractors and with generic observations that permit a perfect embedding, we still have the critical issue of general synchrony (detecting $X \leftrightarrow Y$ could be a false positive in one direction). The problem of synchrony is ultimately a topological one, because the situation that a cross map may be approximately invertible occurs even in the original state spaces.
>
> \[1\] A. McInerney. First Steps in Differential Geometry. Undergraduate Texts in Mathematics. New York, NY: Springer New York. 2013

---

> ### Comment · Reviewer_rcKb · 2024-08-12
> **Thank you**
>
> Thank you for your reply and additional explanations. I will maintain my current score and suggest that the authors add the above explanations to an updated version of the paper.

---

> > ### Author Response · Authors · 2024-08-14
> >
> > We again thank the reviewer for their feedback and engagement with our article.
> >
> > We are grateful for the reviewer's suggestions and are incorporating all of them into the revised manuscript.

---

### Official Review · Reviewer_kcd7 · 2024-07-11

**Soundness:** 2
**Presentation:** 3
**Contribution:** 2
**Rating:** 5
**Confidence:** 3

**Summary:**

This paper proposes a novel tangential space causal inference method, TSCI, for identifying causal relationships from time series data generated by dynamical systems. TSCI considers vector fields as explicit representations of dynamic systems and checks for the degree of synchronization between the vector fields to identify the causal direction. TSCI has higher interpretability and scalability than the original CCM. Experiments on standard systems show that TSCI is much easier to identify causal directions than CCM.

**Strengths:**

1.	This paper innovatively proposes to use tangent vector instead of delay embedding to identify causality in dynamic systems. The correlation between tangent vectors is geometrically motivated, so it has higher interpretability.
2.	TSCI can be combined with any differentiable regression model to learn the cross map, so it is more flexible and can adapt to model agnostic situations.
3.	This paper has a solid theory and clear expression, so readers who lack theoretical knowledge can also understand this article.

**Weaknesses:**

1.	It may be unreasonable to use Pearson correlation coefficient in this paper to evaluate the correlation between tangent vectors. As far as I know, Pearson correlation coefficient can only evaluate linear correlation, if there is nonlinear correlation between tangent vectors, then Pearson correlation coefficient may not work well.
2.	The experiments in this paper are not convincing. It only shows the results of algorithms identifying the causal relationships between a few variables in the toy system. It is recommended to evaluate TSCI in real systems with larger scale data. In addition, the results in this paper are not sufficient to show that TSCI is superior to CCM, because CCM also identities correct causal relationships. The authors should provide results that TSCI can correctly identify causality and CCM cannot.
3.	This paper claims that TSCI is lighter and more scalable than CCM, but lacks the necessary results support.
4.	This paper assumes a high quality of the reconstruction of the latent states, which may not be reasonable. As the article says, a high-quality reconstruction state does not always exist. It is recommended to conduct experiments with low latent state quality, and the robustness of TSCI and CCM for this problem has been compared.
5.	The experiments in this paper lack necessary analysis. For example, the authors should further explain why TSCI is more resistant to the effects of general synchrony than CCM.

**Questions:**

1.	How does TSCI perform on real systems? Can the authors demonstrate the performance of TSCI in identifying causal relationships with large scale data?
2.	Do the authors consider cyclic causal relationships? For example, if there are cyclic causal relationships X->Y->Z->X, can TSCI correctly identify them?
3.	Can TSCI correctly identify causality when the quality of reconstruction of latent states are low?
4.	Did the authors consider nonlinear causality? The Pearson correlation coefficient may not be effective in identifying nonlinear causal relationships.
5.	In Figure 2, why cos(\theta) in the correct direction not equal to -1?

**Limitations:**

The authors do not directly state the limitations in the paper. This work is limited by the quality of reconstruction of latent states, which is a difficult problem to solve, and the authors do not report the robustness of the proposed method for this problem.

---

> ### Author Rebuttal · Authors · 2024-08-07
>
> We thank the reviewer for their time and consideration. We now address the weaknesses and questions mentioned above.
>
> # Weaknesses
> > It may be unreasonable to use Pearson correlation coefficient in this paper to evaluate the correlation between tangent vectors. As far as I know, Pearson correlation coefficient can only evaluate linear correlation, if there is nonlinear correlation between tangent vectors, then Pearson correlation coefficient may not work well.
>
> We address this question in detail in the general response to reviewers. In short, we interpret the correlation used by TSCI as an expected cosine similarity, not Pearson correlation. Furthermore, the linear correlation is only considered after examining tangent spaces (local linearizations). We note that this is well-motivated by the theory of differential geometry; if the cross-map exists, the tangent vectors should align *exactly*.
>
> > The experiments in this paper are not convincing. It only shows the results of algorithms identifying the causal relationships between a few variables in the toy system. It is recommended to evaluate TSCI in real systems with larger scale data.
>
> We are not aware of any real data sets that are freely available online, satisfy the assumptions of Takens' theorem, and have ground truth available. Of the articles that use real data, they are often not available online, or the causal ground truth is unknown (so any differences between TSCI and CCM cannot be interpreted). As a result, it is quite common in the CCM literature to primarily test on synthetic data.
>
> > In addition, the results in this paper are not sufficient to show that TSCI is superior to CCM, because CCM also identities correct causal relationships. The authors should provide results that TSCI can correctly identify causality and CCM cannot.
>
> We disagree that CCM consistently identifies the correct causal relationships in the Rossler-Lorenz system. In Figure 2a, we have plotted the performance of TSCI and CCM for a wide variety of different dynamical systems. We agree that TSCI and CCM provide the same, correct conclusion for $C\lesssim 1.5$. However, for systems with $C\gtrsim 1.5$, a practitioner using CCM would likely (incorrectly) conclude bidirectional causality. On the other hand, TSCI correctly identifies unidirectional causality until $C\approx 2.5$, where the effects of general synchrony are known to occur. Even with $C=1$, it is not clear that CCM consistently performs well, while TSCI clearly separates the two cross map scores (see Figure 2b).
>
> > This paper claims that TSCI is lighter and more scalable than CCM, but lacks the necessary results support.
>
> We believe our claims regarding scalability may have been misunderstood; our claim is not that TSCI is more scalable than CCM, but rather that it enjoys the same scalability and lightweight implementation as CCM. Notably, the only added computational complexity in TSCI is solving a simple least squares problem.
>
> > This paper assumes a high quality of the reconstruction of the latent states, which may not be reasonable. As the article says, a high-quality reconstruction state does not always exist.
>
> See the general response to reviewers for more discussion on this matter. In the manuscript appendix, we consider the effect of poor reconstruction quality in two specific cases: improper selection of the embedding dimension (appendix A.1) and the presence of additive noise in the recordings (appendix A.2).
>
> # Questions
> > How does TSCI perform on real systems? Can the authors demonstrate the performance of TSCI in identifying causal relationships with large scale data?
>
> See the above reply to weakness 2.
>
> > Do the authors consider cyclic causal relationships? For example, if there are cyclic causal relationships X->Y->Z->X, can TSCI correctly identify them?
>
> Theoretically, cyclic causation of the form  $X \rightarrow Y \rightarrow Z \rightarrow X$ is not uniquely identifiable using cross maps. It will likely detect that $X \rightarrow Z$. There are two separate issues for this.
>
> First, CCM-based methods are not designed to handle mediation. This follows by carefully thinking about the motivation for CCM: if $X\to Y$ and $Y\to Z$, then there exist smooth functions $F: \mathcal{M}_Y \to \mathcal{M}_X$ and $G:\mathcal{M}_Z \to \mathcal{M}_Y$. By composition, there exists a function $\mathcal{M}_Z \to \mathcal{M}_X$ as well.
>
> Second, if $X\to Z\to Y$, and $Z\to X$ as well, then $\mathcal{M}_X$ and $\mathcal{M}_Z$ should mutually cross map onto each other. This implies that the manifolds are diffeomorphic, and the direction of causality is indistinguishable.
>
> Note that we would not consider this a weakness of cross map methods. Rather, it is a difference in interpretation. We argue that when interpreting causality in terms of identifying directional coupling, these are the "correct" conclusions.
>
> > Can TSCI correctly identify causality when the quality of reconstruction of latent states are low?
>
> It depends on the type of impurity in the reconstruction process. If vanilla TSCI is directly applied to a noisy reconstruction, it will fail. However, we are able to apply TSCI accurately to noisy data if we apply a  smoothing filter (appendix A.2). Please see our general response to reviewers for more discussion and additional experiments.
>
> > Did the authors consider nonlinear causality? The Pearson correlation coefficient may not be effective in identifying nonlinear causal relationships.
>
> The TSCI method is a nonlinear causal discovery method. Nonlinear mappings between manifolds induce linear mappings between tangent spaces (Lemma 2.1.1). See the general response to reviewers above for discussion on the use of correlation.
>
> > In Figure 2, why cos(\theta) in the correct direction not equal to -1?
>
> In Figure 2, the vectors map onto each other near-perfectly, and the angle $\theta$ between tangent vectors is near 0. Since $\cos(0)=1$, the samples of $\cos(\theta)$ are near 1.

---

> ### Comment · Reviewer_kcd7 · 2024-08-12
> **Response for Rebuttals**
>
> Thank you for your response that addressed most of my concerns, and I have raised my score. But I still have some comments and questions:\
> 1.Although it is difficult to find real datasets that satisfy the assumptions of Taken’ theorem, it is suggested to evaluate TSCI on other general real datasets, especially on larger real datasets, which is conducive to proving the robustness of TSCI.\
> 2.The authors state in abstract that they present a basic TSCI algorithm, which is lightweight and more effective than the basic CCM algorithm, which is contradictory to their response. Is TSCI as lightweight as CCM or lighter than CCM? If TSCI is more lightweight, please give the explanation and experimental results to support this conclusion.

---

> > ### Author Response · Authors · 2024-08-14
> >
> > We thank the reviewer once again for their feedback and engagement with our article. We now reply to the reviewer's remaining comments.
> >
> > (1) The primary issue with finding real data sets is that it is a rather particular scenario to find data which both has a nontrivial causal ground truth as well as having data with some underlying dynamical manifold structure (in the sense of Takens' theorem). While some articles applying CCM and related methods to real data exist, many of these data sets cannot be accessed without permission. Additionally, the vast majority of papers on CCM and related methods overwhelmingly study toy systems and synthetic data sets, because finding clean data with manifold structure is so rare.
> >
> > Of the few articles that use real data sets to test CCM, they typically study data that has no verifiable ground truth, so even in comparison to TSCI we would be unable to verify which method is performing better. There is also somewhat of a survivorship bias present: datasets that CCM has been applied to are ones that CCM performs well on. That said, we have been and will continue to look for real data sets where we can justify the manifold structure assumption.
> >
> > Gernerally, it would be unjustified to apply CCM or TSCI to a data set without manifold structure. To make an analogy, one would not apply a graph neural network to a data set that doesn't have some inherent graph structure. Regardless if the model produces the correct result or not, we have no theoretical guarantee that the result was meaningful unless the structure hypothesis is assumed.
> >
> > One partial solution is to consider more sophisticated synthetic systems. For example, \[1\] considers a large data set of synthetic neural data. Generating synthetic data like this is advantageous because it is reproducible, and the scale of the system can be varied. If this is acceptable, we will incorporate additional experiments of this nature into the revised manuscript.
> >
> > If the reviewer is aware of any potential real data sets that would be appropriate, we will gladly incorporate experiments using them into the manuscript. It is our intention to fairly prove the robustness of the TSCI method, but after searching for some time, we are yet to find real data sets that would provide meaningful comparison between CCM and TSCI.
> >
> > (2) Thank you for specifially pointing out the abstract. While we believe that what we wrote is correct, we understand that it may be easy to misinterpret. In the original abstract, we made two separate claims: (1) TSCI is a lightweight method in general, and (2) TSCI is more effective as an algorithm than CCM. In the revised manscript, we have modified the abstract to better clarify that these two claims are distinct. Specifically, the relevant part now reads:
> > "We first present a basic version of the TSCI algorithm, which is shown to be more effective than the basic CCM algorithm with very little additional computation. We additionally present augmented versions of TSCI that leverage the expressive power of latent variable models and deep learning."
> >
> > Both TSCI and CCM can be implemented with a time complexity of $O(T \log(T))$ using KDTrees, where $T$ is the time series length. This implies that TSCI and CCM have comparable scaling, and both are lightweight.
> >
> > Our claim that TSCI is more effective than CCM is unrelated to the computational complexity, and is based on results of the experiments, where TSCI  outperforms CCM in accuracy.
> >
> >
> >
> > \[1\] E. De Brouwer, et al. "Latent convergent cross mapping." International Conference on Learning Representations (ICLR). 2021.

---

### Official Review · Reviewer_htpT · 2024-07-12

**Soundness:** 4
**Presentation:** 3
**Contribution:** 4
**Rating:** 7
**Confidence:** 3

**Summary:**

The authors propose a novel statistic for detecting causality in dynamical systems, which overcomes a key conceptual difficulty in the convergent cross mapping (CCM) method from Sugihara et al. [2012]. CCM is justified by the existence of a surjective "cross map" from the delay embedding of a driving to a driven dynamical variable in unidirectional coupling. However, the CCM test statistic is difficult to interpret because it does not directly measure whether this map can be constructed.

Instead, the authors represent the dynamical equations through the time invariant flow fields induced by these delay embeddings. A cross map would induce a Jacobian transformation between these flow fields, so the authors propose testing the correlation between the flow fields of a candidate "driven" variable and a candidate "driving" variable pushed forward by the estimated Jacobian.

The authors demonstrate the clear advantage of their method for distinguishing the direction of causality in a unidirectional Rössler-Lorenz system.

**Strengths:**

The authors give a convincing account of the difficulties with CCM and their refinements are sure to be highly impactful to the subject area.

**Weaknesses:**

The authors point out that the CCM statistic does not admit a simple decision rule and refer to their correlation coefficients as test statistics. However, the authors do not offer guidance as to when, e.g., a two-body dynamical system should be considered unidirectionally coupled. The authors could explore testable hypotheses such as "$H_0$: there is no driving $x \Rightarrow y$", perhaps over a limited class of models. Perhaps this has been investigated in the CCM literature already and the authors could adapt existing methods for decision making.

The authors could also address their choice of correlation as their test statistic. Perhaps replacing correlation coefficients with mutual information $I(\mathbf{u};\mathbf{J}_F \mathbf{v})$ should be acknowledged as a possible future refinement in the conclusion (see the first question below).

**Questions:**

How do the authors interpret a measured Jacobian that cannot predict the correct magnitude of $\mathbf{v}(\mathbf{\tilde{y}})$ from $\mathbf{u}(\mathbf{\tilde{x}})$ but correctly predicts the direction? This case, of course, gives the TSCI test statistic value $1$.

How do the authors interpret their results if the measured Jacobian is invertible? Is this simply the case in which $x(t)$ and $y(t)$ are driven by the same set of variables $\mathbf{z}$?

**Limitations:**

Yes. The authors adequately describe underlying assumptions and describe challenges (i.e., generalised synchrony) that obstruct informative results.

---

> ### Author Rebuttal · Authors · 2024-08-07
>
> We thank the reviewer for their time and consideration. We now address the weaknesses and questions mentioned above.
>
> # Weaknesses
> > The authors point out that the CCM statistic does not admit a simple decision rule and refer to their correlation coefficients as test statistics. However, the authors do not offer guidance as to when, e.g., a two-body dynamical system should be considered unidirectionally coupled. The authors could explore testable hypotheses such as $H_0$: "there is no driving", perhaps over a limited class of models. Perhaps this has been investigated in the CCM literature already and the authors could adapt existing methods for decision making.
>
> One approach that we considered for making a decision was to assume that the two vector fields were independent and to apply a uniform distribution over the hypersphere to model their angles. However, we found this approach unsatisfactory in general because it often selected thresholds that were too small to reproduce the ground truth in our experiments. We mostly attribute this to the (sharp) null hypothesis being flawed; the theory of cross-maps does not preclude a weak or non-smooth estimator existing in the causal direction. We therefore do not generally have reason to believe the sharp null hypothesis of independence is realistic. This is also reflected in the mutual information case; the MI between time series with *no* coupling is estimated to be greater than zero, implying some correlation between samples. While MI estimators are not always accurate, they tend to underestimate the MI rather than overestimate it, so if anything, the MI with no coupling is likely to be larger than pictured in Figure 1 of the general response.
>
> The issue of test statistics has been approached several times in the CCM literature, but in our opinion, no single approach is entirely satisfactory. Indeed, one popular approach is to simply set a threshold (say, 0.3) prior to testing. Several other approaches use "surrogates" by shuffling the time series. However, working with these surrogates is delicate, as we cannot easily assume that time series arising from dynamical systems are stationary. Finally, one other approach (for example, used by latent CCM) is to test the difference in the test statistic over simulations where coupling is and isn't present. We argue access to such a surrogate system is overwhelmingly rare in practice, and that this, to a certain extent, begs the question of causal analysis. We will update future versions of the manuscript to include some discussion regarding previous work on decision rules.
>
> > The authors could also address their choice of correlation as their test statistic. Perhaps replacing correlation coefficients with mutual information $I(\mathbf{u}; \mathbf{J}_F \mathbf{v})$ should be acknowledged as a possible future refinement in the conclusion (see the first question below).
>
> Thank you for this interesting and thoughtful suggestion. We ran an experiment to examine how the mutual information behaves compared to the cosine similarity, which will appear in an appendix of the revised manuscript. We defer to our general response to the reviewers above for information and discussion regarding this experiment.
>
> # Questions
> > How do the authors interpret a measured Jacobian that cannot predict the correct magnitude of $\mathbf{v}(\mathbf{\tilde{y}})$ from $\mathbf{u}(\mathbf{\tilde{x}})$ but correctly predicts the direction? This case, of course, gives the TSCI test statistic value 1.
>
> Our intuition is that this situation would be pathological and unlikely to occur in practice. In particular, it would imply that one can learn a cross map, but that the cross map identifies the wrong speed (or magnitude of velocity) consistently across the whole manifold. Generically, we would assume the resulting vector field would not be smooth, which would imply the supposed cross-map also lacked smooth structure (see e.g., Lee, Proposition 8.14). While it is possible that in some places the method over/underestimates the magnitude of the velocity, this is the result of a computational error and cannot be explained from the theoretical development. To check that the magnitude is closely tracked in our experiments, we added Figure 3 to the PDF appearing in the general rebuttal.
>
> > How do the authors interpret their results if the measured Jacobian is invertible? Is this simply the case in which and are driven by the same set of variables?
>
> The Jacobian matrix of the cross map, as a linear map $T_xM \rightarrow T_{F(x)}N$, is invertible whenever the two manifolds in question have the same dimensionality. This is a necessary condition for bidirectional causation, or a common driver, but it is not a sufficient condition. Mathematically, the reason is because an invertible Jacobian $\mathbf{J}_F$ indicates _local_ invertibility of the function $F$, but it does not mean the function is *globally* invertible.
>
> In addition, we expect that invertibility does not commonly occur in the embedded space. This is because we tend to embed in a higher dimension than the actual manifold, in order to ensure Takens' theorem holds. Thus, we would actually expect invertibility to happen on some linear subspace of the tangent plane $T_p M$, reflecting the subspace topology of $M \subset \mathbb{R}^n$.

---

> > ### Comment · Reviewer_htpT · 2024-08-12
> >
> > Many thanks to the authors for their thoughtful responses.
> >
> > The difficulty of decision rules in this case is indeed intriguing!
> >
> > While I remain unconvinced that cosine similarity is an "ultimate" measure of the similarity between $\mathbf{u}$ and $\mathbf{J}_F \mathbf{v}$, I see now that mutual information is an inappropriate suggestion (perhaps a normalised version would fix this). The authors are also right to point out that cosine similarity is more straightforward to estimate than MI.
> >
> > As a somewhat late suggestion, the authors may wish to test an averaged Euclidean distance between the vector fields, which should be computationally tractable and more precisely measures the condition arising from lemma 2.1.1. (which, as far as I can tell, concerns their similarity in both magnitude and direction).
> >
> > On perspectives expressed by reviewers about the scope of this paper, I would argue that the paper being explicitly in conversation with CCM---a popular approach with a very particular notion of "causality"---is a strength rather than a weakness! I feel strongly that this paper is a valuable contribution to the dynamical systems literature and has important and immediate applications. I maintain my score at 7: Accept.

---

> > > ### Author Response · Authors · 2024-08-14
> > >
> > > We thank the reviewer once again for their thoughtful suggestions and positive view of our manuscript.
> > >
> > > We agree that it would be reasonable to try using the Euclidean distance between vector fields. As the reviewer mentioned, this would be consistent with the theory developed in the manuscript, and is straightforward to implement. One disadvantage of this approach, like the MI approach, is that it is unclear what principle should be used to derive a general threshold. However, we found in a simple experiment that there is still a clear difference in the distributions obtained from Euclidean distances in each direction. Our primary preference for cosine similarity arises from (1) its interpretability and (2) the ease of finding heuristics to select a threshold. For example, selecting 0.8 as a desired cosine similarity is about as arbitrary as selecting 80% variance explanation in principal component analysis. In any case, our preference does not preclude the use of alternative measures, such as the Euclidean distance or the MI, and we find these to be very interesting directions for future research.
> > >
> > > In the revised manuscript, we added an appendix section to provide examples of alternative test statistics for the TSCI and to compare them to the cosine similarity. For now, this section includes the MI and the Euclidean distances as suggested by the reviewer.

---

### Official Review · Reviewer_VHEe · 2024-07-13

**Soundness:** 2
**Presentation:** 2
**Contribution:** 2
**Rating:** 4
**Confidence:** 3

**Summary:**

The authors propose the Tangent Space Causal Inference (TSCI) method for detecting causalities in dynamic systems. TSCI works by considering vector fields as explicit representations of the systems’ dynamics and checks for the degree of synchronization between the learned vector fields. The authors present both a basic TSCI algorithm, which is lightweight and more effective than the basic CCM algorithm, as well as augmented versions of TSCI that leverage the expressive power of latent variable models and deep learning. The authors demonstrate improved causal inference performance across a number of benchmarks.

**Strengths:**

The method leverages vector fields of manifolds and improve the CCM method for causal direction detection between time series. Applying tangent space vector representation  for causal discovery is interesting. The author also provides theoretical analysis to support the algorithm.

**Weaknesses:**

a. The main weakness of the paper is that the experiments are insufficient to validate the method. Comparison with existing standard methods is missing. Experimental comparisons with classical Granger causal discovery and other methods are necessary.

b. Theoretical analysis and comparison with existing mechanisms and methods should be included.

c. Apart from comparison with classical time series causal discovery methods, comparison with bivariate causal discovery should be included to strengthen the paper. E.g. papers [1-7]


[1]Blöbaum, P., Janzing, D., Washio, T., Shimizu, S. and Schölkopf, B., 2018, March. Cause-effect inference by comparing regression errors. In International Conference on Artificial Intelligence and Statistics (pp. 900-909). PMLR.


[2]Khemakhem, I., Monti, R., Leech, R. and Hyvarinen, A., 2021, March. Causal autoregressive flows. In International conference on artificial intelligence and statistics (pp. 3520-3528). PMLR.

[3]Ren, S. and Li, P., 2022, October. Flow-based perturbation for cause-effect inference. In Proceedings of the 31st ACM International Conference on Information & Knowledge Management (pp. 1706-1715).

[4]Daniusis, P., Janzing, D., Mooij, J., Zscheischler, J., Steudel, B., Zhang, K. and Schölkopf, B., 2012. Inferring deterministic causal relations. arXiv preprint arXiv:1203.3475.

[5]Fonollosa, J.A., 2019. Conditional distribution variability measures for causality detection. Cause Effect Pairs in Machine Learning, pp.339-347.

[6]Ren, S., Yin, H., Sun, M. and Li, P., 2021, December. Causal discovery with flow-based conditional density estimation. In 2021 IEEE International Conference on Data Mining (ICDM) (pp. 1300-1305). IEEE.

[7]Hoyer, P., Janzing, D., Mooij, J.M., Peters, J. and Schölkopf, B., 2008. Nonlinear causal discovery with additive noise models. Advances in neural information processing systems, 21.

**Questions:**

What are the advantages of using corr(xˆ(t), x(t)) compared to Granger causal discovery, and the regression or conditional variance-based methods[1,3,6]?

**Limitations:**

Experimental study and comparison with additional baselines should be included to support the proposed method.

---

> ### Author Rebuttal · Authors · 2024-08-07
>
> We thank the reviewer for their time and consideration. We now address the weaknesses and questions mentioned above.
>
> # Weaknesses
> > The main weakness of the paper is that the experiments are insufficient to validate the method. Comparison with existing standard methods is missing. Experimental comparisons with classical Granger causal discovery and other methods are necessary.
>
> Comparisons to the methods Granger causality, RECI, IGCI, and ANM have been added to our experiments. We refer to the general response to reviewers above for more information and for the test results.
>
> > Theoretical analysis and comparison with existing mechanisms and methods should be included.
>
> We include theoretical analysis to support the method in the main manuscript. We have added more theoretical comparisons to Granger causality in the general response and a new appendix, but we also feel that comparisons between cross map-based methods and traditional causal discovery methods appear exhaustively in the CCM literature, for example, in [1-3].
>
> > Apart from comparison with classical time series causal discovery methods, comparison with bivariate causal discovery should be included to strengthen the paper. E.g. papers [1-7]
>
> Comparisons to the methods RECI, IGCI, and ANM have been added to our experiments. We refer to the general response to reviewers above for more information and for the test results. We found that these methods performed underwhelmingly compared to CCM and TSCI, due to the nature of the systems under study.
>
> # Questions
> > What are the advantages of using corr(xˆ(t), x(t)) compared to Granger causal discovery, and the regression or conditional variance-based methods[1,3,6]?
>
> CCM and TSCI are intended to be used to study signals generated by coupled, deterministic dynamical systems. In this setting, Granger causal discovery fails because the separability condition is violated. We refer to the general response to reviewers above for more discussion on the matter.
>
> Just as the main assumption of Granger causality fails in coupled deterministic dynamical systems, the underlying assumptions of most bivariate causal discovery methods are also violated.The conditional variance based methods [1,3,6] are not appropriate because we typically do not expect an additive noise model. For example, from A2 of [3], we would not generally expect an invertible, monotonic function such that $g(x) = y$ to exist. This is apparent by looking at a scatter plot of $X$ and $Y$. Indeed, we wouldn't even expect a *function* to exist in general --- please see Figure 4 of the PDF attached to our general response. Such assumptions are typical in bivariate causal inference, for example in ANM or IGCI, which is why we chose to omit comparisons in the original manuscript. We verified that RECI, IGCI and ANM failed in the additional experiment described in the general response to reviewers above.
>
> # References
> \[1\] G. Sugihara, R. May, H. Ye, C. Hsieh, E. Deyle, M. Fogarty, and  S. Munch. Detecting causality in complex ecosystems. Science, 338(6106):496–500, 2012.
>
> \[2\] E. De Brouwer, A. Arany, J. Simm, and Y. Moreau. Latent convergent cross mapping. In International Conference on Learning Representations, 2020.
>
> \[3\] Assaad, C. K., Devijver, E., and Gaussier, E. Survey and evaluation of causal discovery methods for time series. Journal of Artificial Intelligence Research, 73, 767-819. 2022.

---

> ### Comment · Reviewer_VHEe · 2024-08-11
>
> Thank you for the response.  The authors failed to provide a sufficiently direct and convincing comparison between bivariate causal discovery methods and the proposed method, TSCI.  Though the proposed TSCI is targeting causal discovery in dynamic systems, with powerful backbone models, e.g. neural networks, I believe the bivariate causal discovery methods, such as RECI[1], and the deep generative model-based methods CAREFL[2], EFRE[3], retain the power to capture the causal relationship between dynamic systems.  The authors need to provide a comprehensive study to compare these methods using both dynamics systems data and real-world bivariate causal discovery datasets, e.g. Tuebingen dataset.
>
> The theoretical advantage of the proposed method is not clear to the reviewer.  Existing bivariate causal methods may also be able to be applied to the case X->Y, where -> is a dynamic system or process.  The authors should provide convincing experimental study and analysis, and theoretical analysis to demonstrate the advantage of the proposed method.
>
> Therefore, I will keep the score unchanged.
>
> [1]Blöbaum, P., Janzing, D., Washio, T., Shimizu, S. and Schölkopf, B., 2018, March. Cause-effect inference by comparing regression errors. In International Conference on Artificial Intelligence and Statistics (pp. 900-909). PMLR.
>
> [2]Khemakhem, I., Monti, R., Leech, R. and Hyvarinen, A., 2021, March. Causal autoregressive flows. In International conference on artificial intelligence and statistics (pp. 3520-3528). PMLR.
>
> [3]Ren, S. and Li, P., 2022, October. Flow-based perturbation for cause-effect inference. In Proceedings of the 31st ACM International Conference on Information & Knowledge Management (pp. 1706-1715).

---

> > ### Author Response · Authors · 2024-08-14
> >
> > We thank the reviewer for their response.
> >
> > We repeat that the application of TSCI to arbitrary bivariate causal discovery data sets is unjustified. Without assuming an underlying dynamical manifold structure, there is no theoretical basis as to why CCM or TSCI should yield the correct causal truth. This is entirely analogous to the invalidity of RECI/IGCI/EFRE when no deterministic function $y = f(x)$ exists. It is unclear what the scientific value of applying TSCI to a bivariate causal discovery data set (e.g., the Tuebingen data) would be when there is no inherent time series structure, and the TSCI method is specifically intended for time series generated by dynamical systems.
> >
> > The theoretical advantage of TSCI and CCM, as cross mapping methods, is that they address scenarios that may be considered pathological by traditional causal discovery methods. It is well known that the separability assumption of Granger causality is violated in deterministic dynamical systems. Static causal discovery methods cannot account for spurious correlations that may occur due to autocorrelation or non-stationarity in time series data. Furthermore, static methods such as RECI, IGCI, and EFRE, make specific assumptions about the existence of a deterministic function $y=f(x)$, which are clearly violated in the time series we are interested in.
> >
> > In the PDF attached to the main author rebuttal above, we also show empirically Granger causality, RECI, IGCI, and ANM fail in the Rossler-Lorenz system. This is not a criticism of these methods, but rather a reminder that TSCI and CCM are methods specifically designed to addressthe blind spots of many traditional causal discovery techniques.

---

### Author Rebuttal · Authors · 2024-08-07

We would like to thank the reviewers for their thoughtful comments. We address some common concerns and questions here.

# Comparisons to Granger Causality and Other Methodologies
As mentioned in the introduction, Granger causality (GC) can perform poorly on signals generated by deterministic dynamical systems. GC assumes separability, i.e., that novel probabilistic information about the future is continually generated by the cause, and this information is unique to the cause. Thus, GC is most readily applied to stochastic systems. In contrast,  deterministic dynamical systems violate separability due to the presence of deterministic rules for the system's evolution. This same violation ultimately makes the anti-causal prediction in the CCM/TSCI approaches possible: the cross-map exists because the cause leaves such a strong signature on the effect that recovering the dynamical rule for the system is possible (a consequence of Takens' theorem, see \[5\]). As a result, CCM/TSCI is often applicable specifically because the assumptions of GC are violated.

We expanded our comments in the introduction and included an appendix to better explain comparisons to GC. As this is relatively well-known in the CCM literature, we also leave several references to past discussions, e.g., the appendices of \[1\] and \[2\]. We added a comparison to GC in our Rossler-Lorenz system, showing that GC always incorrectly predicts bidirectional causality for $C > 0$ (Table 1 of the attached PDF).

At the suggestion of Reviewer VHEe, we added additional comparisons to the methods RECI, IGCI, and ANM. These methods are expected to perform quite poorly as several of their assumptions are harshly violated. Namely, even in the situation that $X \rightarrow Y$, we would not expect that $y_t = f(x_t)$ (see e.g. Figure 4 attached), and we would especially not expect an additive noise decomposition. Our results are in Tables 1, 2, 3 in the attached PDF, and are summarized as follows: IGCI consistently fails to detect a causal edge; RECI consistently chooses the direction $X \rightarrow Y$, but does so even when $C=0$, suggesting this result is *not* actually detecting causality -- to this end, the variance ratios are typically large; finally, for all $C > 0$, ANM detects bidirectional causality.

# On the Use of Cosine Similarity, and Alternative Measures
We believe the TSCI score is best understood as a cosine similarity, not Pearson correlation. This is only computationally equivalent because vectors in the tangent space are centered. This method doesn't preclude nonlinear relationships, since tangent vectors are measured locally. By Lemma 2.1.1 in the manuscript, every smooth function $F$ induces a linear mapping $\mathbf{J}_F$ between tangent spaces which aligns the systems' vector fields.

Reviewer htpT suggested the mutual information $I(\mathbf{u}; \mathbf{J}_F \mathbf{v})$ as a possible alternative test statistic. Though lacking direct justification via Lemma 2.1.1, the use of MI seems intuitive, and in Figure 2 of the attached PDF we test MI and cosine similarity on the Rossler-Lorenz test systems. We find that MI typically shows less separation than TSCI. Two other issues are difficulties interpreting any particular value of MI, and that estimating MI from samples is a difficult, non-trivial problem.

# Comments on Existing Experiments
We believe several experiments in the appendices of our manuscript partially address reviewers' concerns. The experiment in Appendix A.1 demonstrates how poor manifold reconstruction, due to different selections of embedding dimensions, affects the resulting performance. The experiment in Appendix A.2 similarly considers the effect of additive noise in the observed signals. We recognize that these experiments should be better referenced in the main manuscript, and added text to the main manuscript explaining additional experiments, including the new experiments conducted during this rebuttal period.

# Comments on the Quality of Reconstructions
There are many ways to achieve a poor state space reconstruction, and we therefore cannot make general statements about robustness to reconstruction quality. In the appendix of the manuscript, we consider poor reconstructions due to poor selection of the embedding dimension (A.1) and additive noise (A.2). In these experiments, both CCM and TSCI behave similarly and in a manner consistent with the theory.

Generally, there are many non-trivial ways in which the state reconstruction can be poor, e.g., presence of periodic trends \[3\] and underexploration of the shadow manifold \[4\]. A systematic study of how TSCI depends on the reconstruction quality would already provide ample material for future work. We considered one additional experiment to study how CCM and TSCI respond to a dynamical confounder in the form of an added sine wave. We visualize our results in Figure 3 of the attached PDF. These results suggest that TSCI may be robust to a moderate confouding influence, deteriorating only when the signal power of the confounder overtakes the signal power of the signals of interest. Notably, TSCI seems significantly more robust to false claims of strong causation when the relative power of the confounder is large.


# References
\[1\] G. Sugihara, R. May, H. Ye, C. Hsieh, E. Deyle, M. Fogarty, and  S. Munch. Detecting causality in complex ecosystems. Science, 338(6106):496–500, 2012.

\[2\] E. De Brouwer, A. Arany, J. Simm, and Y. Moreau. Latent convergent cross mapping. In International Conference on Learning Representations, 2020.

\[3\] S. Cobey, and E.B. Baskerville. Limits to Causal Inference with State-Space Reconstruction for Infectious Disease. PLoS One. 2016

\[4\] K. Butler, G. Feng, and P. M. Djuric. On causal discovery with convergent cross mapping. IEEE Transactions on Signal Processing, 2023.

\[5\] J. Stark. Delay embeddings for forced systems. I. Deterministic forcing. Journal of Nonlinear Science, 9, 255-332. 1999.

---

### Decision · Program_Chairs · 2024-09-25

**Decision:**

Accept (poster)

**Comment:**

This submission was found to tackle an important problem that receives growing attention, clearly highlights potential pitfalls and difficulties with CCM, and develops a novel algorithmic idea to determine causality in dynamical systems. It was found to be clearly written and well structured such that it is mostly easy to read and follow. I do not believe that a comparison with "static" bivariate causal discovery algorithms is required and agree with the authors that no meaningful contributions can be drawn from such a comparison. I encourage the authors to please add their main clarifying arguments from the rebuttal to the camera ready version and to potentially also think about and address some of the minor remaining concerns of the reviewers.